# REFERENCE-FREE RATING OF LLM RESPONSES VIA LATENT INFORMATION

## ABSTRACT

How reliable are single-response LLM-as-a-judge ratings without references, and can we obtain fine-grained, deterministic scores in this setting? We study the common practice of asking a judge model to assign Likert-scale scores to free-text responses and show two systematic issues: scores are unstable under sampling and poorly calibrated, leading to compression near the top of the scale and frequent ties. We then propose and evaluate *Latent Judges*, which derive scalar ratings from internal model signals: (i) probability-weighted scores over integer ratings, (ii) verifier-style probabilities of "yes", and (iii) linear probes trained on model activations at the rating position. Across a broad suite of pairwise and single-rating benchmarks, latent methods match or surpass standard prompting, with consistent gains on pairwise accuracy and listwise ranking relevant to Best-of-$N$ selection. Probability-weighted scores achieve the strongest single-rating correlations, while probes recover useful signals when output logits are miscalibrated. These results indicate that latent information provides deterministic and more discriminative signals for reference-free evaluation, and can improve selection and training approaches like Best-of-$N$, multi-teacher distillation, and routing.

## 1 INTRODUCTION

Different Large Language Models (LLMs) have distinct strengths and weaknesses, and even a single model can produce responses of varying quality to the same prompt. This variability has been productively exploited by Best-of-$N$ sampling at inference (Cobbe et al., 2021; Zhang et al., 2025a) and by post-training with Group Relative Policy Optimization (GRPO) (Guo et al., 2025). It also enables routing, which selects the right model for a given input (Ong et al., 2025; Zhang et al., 2025b), and multi-teacher distillation, which transfers knowledge across models (Timiryasov & Tastet, 2023; Roth et al., 2024; Tian et al., 2025; Gu et al., 2025). Across these settings, we often need *reference-free* judgements of response quality that are *fine-grained* and, ideally, *deterministic*.

Much of the prior work has focused on *verifiable* settings, such as code, math, factuality, or formatting, where correctness can be checked objectively (Guo et al., 2025; Zhang et al., 2025a). In contrast, evaluating the quality of responses to arbitrary prompts is harder (Gehrmann et al., 2021). Here, the prevailing approach is the LLM-as-a-Judge paradigm (Zheng et al., 2023), in which a model rates (or compares) responses, typically on a 5-point Likert scale (Lambert et al., 2024; Hashemi et al., 2024; Lee et al., 2024).

However, our analysis reveals two important issues when obtaining *single-response*, *reference-free* ratings via prompting. First, unless decoding greedily, scores are *unstable*: the same response can receive different ratings across runs. Second, ratings are often *poorly calibrated*: they compress near the top of the scale, leading to frequent ties and limited discriminability. These problems arise from generating discrete tokens on bounded scales under stochastic decoding, and they persist even with strong judge models.

These limitations matter in practice. Reward learning benefits from *fine-grained, unconstrained* scalar signals that better reflect true quality and are more robust to distractors (Tripathi et al., 2025). Best-of-$N$ selection (Cobbe et al., 2021; Lightman et al., 2023) and multi-teacher distillation require clear, tie-resistant rankings *without* reference answers for calibration. Likewise, routing needs consistent per-response scores to choose among models. In short, many high-impact applications require *deterministic and discriminative* reference-free evaluation.

To address this, we propose and evaluate *Latent Judges*, which derive scalar ratings from internal model signals instead of only from generated tokens. We study three complementary families: (i) *probability-weighted ratings*, which compute the expectation over integer scores using the model's next-token distribution; (ii) *verifier-style ratings*, which use the probability of "yes" in a binary "is this response good?" query; and (iii) *latent probes*, lightweight classifiers trained on hidden activations at the rating position. These methods expose *latent information* that is inherently real-valued and deterministic (for fixed inputs), can be rescaled to address calibration, and can recover useful quality signals even when output logits are miscalibrated.

In summary, our contributions are: (1) We systematically evaluate weaknesses of ordinal, single-response LLM-as-a-judge ratings, i.e. instability, compression, and ties, across a wide range of general and finetuned judge models. (2) We propose to mitigate these limitations by using *latent* model information (probabilities and probes) to produce deterministic, fine-grained ratings. (3) We demonstrate that across standard LLM-as-a-judge metrics, latent judges perform on par with or better than prompting baselines. (4) We show how these insights improve practically relevant applications, including listwise ranking for Best-of-$N$ selection and the design of LLM routers.

## 2 RELATED WORK

**LLM-as-a-Judge.** LLM-as-a-Judge has emerged as a practical alternative to traditional human and metric-based evaluation, with models such as GPT-4 shown to align closely with human judgments (Zheng et al., 2023; Li et al., 2024). This paradigm has been applied to holistic quality scoring (Kim et al., 2024a), pairwise preference comparisons (Zheng et al., 2023), and multi-rubric assessments (Lee et al., 2024; Hashemi et al., 2024), as well as domain-specific tasks like medical text generation (Brake & Schaaf, 2024), legal reasoning (Ryu et al., 2023), and financial analysis (Xie et al., 2023).

Recent work has sought to improve the reliability and faithfulness of evaluation through *prompt-based methods* (e.g., GPTScore (Fu et al., 2024), G-Eval (Liu et al., 2023)), which guide evaluation with optimized prompts, explicit rubrics, or reasoning. *Fine-tuned judges* such as JudgeLM (Zhu et al., 2025), Auto-J (Li et al., 2023), Prometheus (Kim et al., 2024a;b), and Themis (Hu et al., 2025a) directly adapt models to human preferences. However, this requires specialized data collection and model training and may not generalize well beyond the training set (Huang et al., 2024).

Finally, adapting LLM-as-a-judge to settings beyond pairwise comparison, which is relevant in practice, is challenging. This is because of the inherent limitations of Likert scale ratings, a point we demonstrate extensively in this paper. Moreover, pairwise comparisons scale poorly to listwise rankings due to context limits, order bias, and non-transitivity (Xu et al., 2025; Hu et al., 2025b).

**Verifiers.** To evaluate correctness in domains where objective correctness of responses can be determined, like math reasoning or code generation, training verifiers is successful (Cobbe et al., 2021; Uesato et al., 2022; Wang et al., 2023; Lightman et al., 2023; Yu et al., 2024; Luo et al., 2024; Hosseini et al., 2024). Verifiers classify responses as correct or incorrect, which serves as a reward model and helps Best-of-$N$ selection (Cobbe et al., 2021; Zhang et al., 2025a). Beyond trained verifiers, having a binary classifier assess the degree of correctness via the probability of "yes" has found application in math/coding (Zhang et al., 2025a) and prompt following evaluation in text-to-image generation (Lin et al., 2024). Our work extends this method to arbitrary natural language generation, where reference answers may not exist and evaluation requires going beyond binary correctness.

**Reward Models.** Scalar reward models are fine-tuned, typically smaller LLMs trained to predict a single scalar score that assesses the quality of a response given a prompt (Christiano et al., 2017; Stiennon et al., 2020; Rafailov et al., 2023). Although reward models share our use of scalar scoring and training, they diverge significantly in that they require training pairs on a large scale. For example, Skywork V2 (Liu et al., 2025) uses 26 million pairs, whereas our latent probes utilize only a few thousand. Unlike our latent probe approach, fully training the reward model sacrifices the ability to query for rationales and may degrade the reasoning capabilities needed for complex judgments. Leaving the base LLM unchanged preserves essential prompt robustness and generalization, which is preferable as specifically trained judge models show only negligible performance advantages.

**Latent Probing.** LLMs encode rich information about the input text in their latent activations (Peters et al., 2018; Devlin et al., 2019). Many works have explored how to extract this knowledge using lightweight classifiers, i.e. *probes* (Veldhoen et al., 2016; Ettinger et al., 2016; Alain & Bengio,

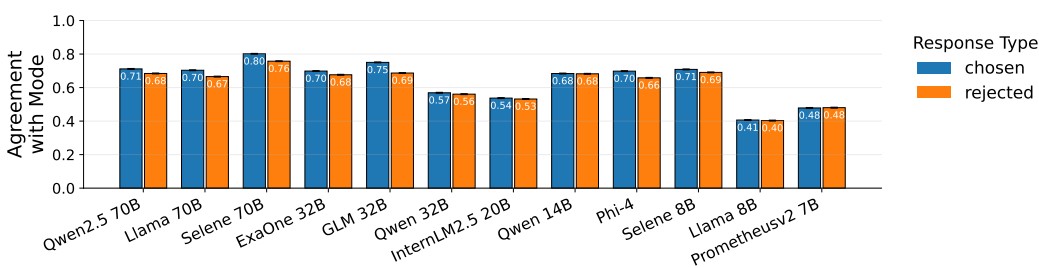

Figure 1: Agreement of ratings with the mode across 10 sampled runs. Even the most consistent models fall below 80% agreement, and some near 40%.

2017; Adi et al., 2017; Conneau et al., 2018). These representations become even more informative when combined with the strong reasoning capabilities of LLMs through targeted prompting (Zou et al., 2023; Marks & Tegmark, 2024; Yang et al., 2024), which enables applications such as steering (Turner et al., 2023; Rimsky et al., 2024), hallucination detection (Obeso et al., 2025; Orgad et al., 2025), and detecting true or false statements (Marks & Tegmark, 2024; Maiya et al., 2025). Building on this, we explore probing as robust text evaluation. Specifically, we extract logits from judge LLMs and train probing classifiers on their internal activations to rate a response, which transforms these latent signals into stable and fine-grained ratings, overcoming the key shortcomings of existing LLM-as-a-Judge methods. While latent information is used for other tasks as detailed above, we systematically evaluate it for rating responses. This changes the evaluation paradigm from ratings as text outputs of the judge LLM to the use of the rich information judge LLMs encode internally.

## 3 Limitations of LLM-as-a-Judge and How to Fix Them

### 3.1 Uncovering Weaknesses of LLM-as-a-Judge Approaches

We systematically examine the limitations of using LLMs to assign holistic ratings to single responses without a reference. This setting is realistic and important, for example, for Best-of-$N$ selection and multi-teacher distillation, but our analysis reveals two weaknesses: ratings are inconsistent across runs and poorly calibrated. As a result, scores fluctuate with the decoding seed and concentrate near the top of the scale, limiting their ability to distinguish response quality.

**Data and Models.** Our experiments use the Tülu Preference Mixture dataset (Lambert et al., 2024), which contains 273,000 prompts paired with chosen and rejected responses. We sample 5,000 prompts and their responses to evaluate 12 judge models, including three trained specifically as judges: Prometheus-v2 8B (Kim et al., 2024b), Selene-1 Mini 8B, and Selene-1 70B (Alexandru et al., 2025). We also include models of varying sizes, from 7B to 70B parameters. Models are prompted with two variants of the Prometheus template (Kim et al., 2024a;b): one using a 1–5 scale and another using a 1–10 scale. The concrete prompts are shown in Appendix C.1. Ratings are generated with temperature 0.7, with 10 scores per response sampled using different seeds. This setup allows us to assess variability and calibration. Since temperature impacts consistency, we ablate its effects in Appendix A.1, confirming our observations for a range of temperature values.

**Finding: LLM judges are inconsistent.** Figure 1 reports the agreement of individual ratings with the mode across 10 samples. Even the most stable models, such as Qwen2.5 70B and Selene 70B, reach only 70–80% agreement. Others, including Prometheus-v2 7B and Llama 8B, drop to 40–50%. Thus, one in five ratings, and often more, diverges from the most frequent score. Some variation is expected under stochastic decoding, but this level undermines reliability. Because scores cluster in a narrow range, even small inconsistencies have a large impact.

**Finding: LLM ratings are not calibrated.** Calibration is another issue. Figure 2 shows that chosen responses nearly always receive high scores (often above 8 on a 1–10 scale), while rejected responses are only slightly lower. For instance, Qwen2.5 70B and Selene-1 assign averages between 7 and 8 even to rejected responses. This compression obscures meaningful differences: without a reference, models default to generous ratings because most responses seem strong in isolation.

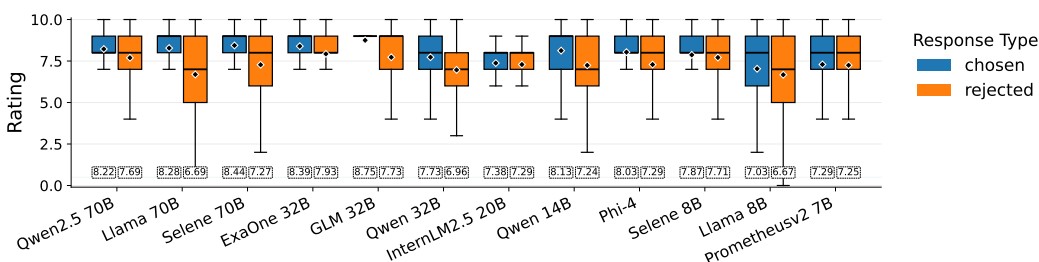

Figure 2: Average ratings for chosen and rejected responses across judge models. Scores are high overall and differences are small.

| | Model | Qwen2.5 70B | Llama3 70B | Selene 70B | ExaOne3.5 32B | GLM4 32B | Qwen3 32B | Int. LM2.5 20B | Qwen3 14B | Phi-4 14B | Selene 8B | Llama3 8B | Prom.v2 7B |
|---|---|---|---|---|---|---|---|---|---|---|---|---|---|
| | | | | | | Comparison to GPT-4 Ratings | | | | | | | |
| 5 | Strict | 35.7 | 50.1 | 50.0 | 29.1 | 39.8 | 37.1 | 16.9 | 40.9 | 29.4 | 23.3 | 32.5 | 20.6 |
| | Lenient | 87.5 | 92.6 | 93.8 | 86.2 | 91.3 | 88.5 | 85.9 | 88.0 | 91.6 | 82.2 | 75.3 | 80.9 |
| 10 | Strict | 38.9 | 61.2 | 56.4 | 32.9 | 44.6 | 49.8 | 34.3 | 49.9 | 42.7 | 30.2 | 42.6 | 34.6 |
| | Lenient | 83.1 | 88.3 | 90.4 | 80.6 | 88.2 | 77.7 | 68.1 | 80.8 | 82.4 | 75.2 | 65.5 | 65.3 |
| | | | | | | Comparison to Skywork-Reward-V2 Ratings | | | | | | | |
| 5 | Strict | 36.8 | 51.0 | 51.2 | 30.3 | 40.3 | 38.3 | 18.3 | 42.5 | 30.3 | 25.9 | 34.3 | 22.0 |
| | Lenient | 88.6 | 93.5 | 95.0 | 87.4 | 91.8 | 89.8 | 87.3 | 89.6 | 92.6 | 84.7 | 77.1 | 82.3 |
| 10 | Strict | 40.7 | 62.5 | 58.2 | 34.7 | 45.2 | 51.9 | 36.3 | 51.8 | 44.6 | 33.3 | 44.7 | 36.3 |
| | Lenient | 84.9 | 89.5 | 92.2 | 82.4 | 88.9 | 79.7 | 70.1 | 82.8 | 84.4 | 78.3 | 67.6 | 67.0 |

Table 1: Agreement of judge models with GPT-4 preferences and Skywork-Reward-V2 preferences. Strict metrics require chosen responses to score higher than rejected ones; lenient metrics also allow ties. Large gaps indicate poor discriminability of single-response ratings.

High scores also lead to frequent ties. Table 1 compares strict and lenient agreement with GPT-4 preferences and additionally with ratings by `Skywork-Reward-V2-Llama-3.1-8B` (Liu et al., 2025), a state-of-the-art reward model. Strict agreement, requiring chosen responses to score higher, is low across all models (e.g., 50.1% for Llama 70B and 16.9% for InternLM2.5 20B on the 1–5 scale). Lenient agreement, which counts ties as correct, is much higher (e.g., 92.6% and 85.9% respectively). The large gaps show that ties are pervasive and that single-response ratings lack discriminative power.

**Discussion of Findings.** In our experiments, we find instability of LLM ratings and insufficient calibrations. The instability under sampling is due to increased epistemic uncertainty of the judge model, i.e. the model is not sure about the precise rating. When forced to unambiguously categorize preferences and scores in discrete rubrics, this excess uncertainty leads to increased entropy in predicted score distributions, which directly explains the instability under stochastic decoding. We attribute issues with calibration to an overfocus on linguistic quality, i.e. the model reserves the lower half of the scale to responses that are ungrammatical or semantically nonsensical. However, this assumption is not informative for modern LLMs which generally give fluent and relevant answers, whose quality may however still substantially differ depending on the prompt.

**Summary.** Overall, the holistic scoring of single responses without references is unreliable. Ratings vary substantially across runs, with even the best models disagreeing one-quarter of the time. Scores are also inflated and compressed near the top of the scale, producing frequent ties that obscure differences. While pairwise prompts can mitigate this when both responses are shown together, our analysis highlights fundamental weaknesses of single-response holistic scoring, which is the focus of this work. These findings motivate us to look for methods that mitigate these issues by providing deterministic and discriminative ratings.

## 3.2 USING SCORES DERIVED FROM LATENT INFORMATION TO RATE RESPONSES

As shown in Section 3.1, LLM-as-a-judge for single responses has several drawbacks. The ratings are often unstable, saturated near the top of the scale, and lack discriminative power. These is-

sues stem from using discrete categories, bounded scales, and stochastic decoding (Holtzman et al., 2020). We investigate to what extent extracting the latent knowledge of a Judge LLM immediately after it processes the prompt can mitigate these issues. This approach offers, in theory, clear advantages: it is deterministic because it does not rely on sampling the LLM's output, and it is discriminative because its scores are real-valued instead of integers on an ordinal scale. Furthermore, these real-valued scores can be scaled and shifted to solve potential calibration issues.

To extract latent knowledge, we investigate three different methods. In *probability-weighted ratings*, we calculate a weighted average of integer ratings. We prompt the LLM to rate a response on a scale, such as 1 to 10, and to output only the rating. Then, we take the token probabilities for the next predicted token, extract the probabilities that correspond to the integers on the scale, and compute a weighted average. This is formally expressed as:

$$S_p(\text{prompt}) = \sum_{i=1}^{n} n \times p_{\text{LLM}}(n \mid \text{prompt}) \tag{1}$$

In *binary* or *verifier-style ratings*, we ask the LLM if the given response is good for the given prompt, and the LLM should only output either "yes" or "no". We then use the probability of predicting "yes" as the rating, formally:

$$S_b(\text{prompt}) = p_{\text{LLM}}(\text{yes} \mid \text{prompt}) \tag{2}$$

This approach is common in mathematical and coding settings, where an objective ground truth exists. In these contexts, models are trained to predict whether a given response is correct. Our innovation transfers this setup to general-purpose prompts, where the notion of "correctness" does not exist. Inspired by encouraging results from evaluating prompt-image correspondence in synthetic images (Lin et al., 2024), we use base LLMs to judge whether a response is good ($P(\text{yes}) = 1$). Unlike typical verifier models, these base LLMs are not fine-tuned.

Finally, in *latent probing*, we train linear probes on the latent activations of a Judge LLM. Specifically, we extract the residual stream activation at the position of the next predicted token after prompting the model to evaluate a response. This option is particularly useful when the LLM's logits are not well-calibrated and may not faithfully reflect its internal states. In these cases, training probes enable better control over the resulting score distribution. Formally, for a Judge LLM $f_\theta$, given an input sequence of length $T$, we extract activation $z^{(l)} \in \mathbb{R}^d$ from layer $l$ at position $T+1$. In this paper, we use activations from intermediate layers (e.g., layer 30 of 40 for Phi-4, see Appendix B and Fig. 12 for the justification). We then train a lightweight probe $g_\phi : \mathbb{R}^d \to \mathbb{R}$ that predicts a quality score $\hat{y} = g_\phi(z^{(l)})$. Probes can be linear or small MLPs. Details on how we train probes on activations are in Appendix B.

## 4 EXPERIMENTAL EVALUATION OF LLM-AS-A-JUDGE BENCHMARKS

### 4.1 PAIRWISE AND SINGLE-RATING BENCHMARKS

We evaluate models on two types of benchmarks: *Pairwise* and *Single-Rating*. These are the traditional baselines to evaluate LLM Judges (Kim et al., 2024a;b; Alexandru et al., 2025). In *pairwise* benchmarks, we assess if the LLM Judge assigns a higher score to a response preferred by humans or GPT-4 than to its dispreferred counterpart. In *single-rating* benchmarks, we evaluate if the scores correlate with ground-truth ratings from human raters or GPT-4.

**Pairwise Benchmarks.** We evaluate models on the following benchmarks: MT-Bench Human Preferences (Zheng et al., 2023), RewardBench (Lambert et al., 2025), HHH-Alignment (Askell et al., 2021), RewardBench-2 (Malik et al., 2025), Auto-J (Li et al., 2023), LFQA (Xu et al., 2023), PreferenceBench (Kim et al., 2024a), and JudgeBench (Tan et al., 2025). All benchmarks consist of ⟨prompt, chosen, rejected⟩ triplets. We only consider unambiguous preferences, where there is a chosen and a rejected response, and we exclude ties, as modeling ties is not the focus of this paper.

For MT-Bench and HHH-Alignment, we use deduplicated versions from (Kim et al., 2024a), removing all repeated ⟨prompt, chosen, rejected⟩ triplets. We also include 10,000-sample subsets from the Tülu Preference Mixture (Lambert et al., 2024) and UltraFeedback (Cui et al., 2024) as additional evaluation sets. The standard evaluation metric for these benchmarks is accuracy, which measures the rate at which predicted rankings agree with the ground truth.

**Single-Rating Benchmarks.** We include the following single-rating benchmarks: FLASK (Ye et al., 2024), MTBench (Zheng et al., 2023), Vicuna-Eval (Kim et al., 2024a), BiGGen Bench (Kim et al., 2025), and UltraFeedback (Cui et al., 2024). For FLASK and BiGGen Bench, both human and GPT-4 ratings are available. For UltraFeedback, we use only the rejected responses and their scores from the binarized version because the score distribution is skewed towards the max score. Benchmarks vary in size, from 320 samples for Vicuna-Eval and MT-Bench to 68,805 for the GPT ratings in BiGGen Bench. All ratings are on a scale from 1 to 5, and the mean scores are positively skewed (i.e., above 3) in all benchmarks. This means that, as all benchmarks resemble the 5-scale baseline and skew positive, this favors the baseline methods.

## 4.2 LLM MODELS AND BASELINES

We compare the rating methods described in Section 3.2 to LLM Judges using the same 5-scale and 10-scale prompts evaluated in Section 3.1, as this is our main point of comparison. Since this baseline produces frequent ties, as shown in Section 3.1, we ensure a fair comparison by breaking ties randomly. For baselines, probability-weighted ratings, and verifier-style ratings, we use the following models: Phi-4 (Abdin et al., 2024), Qwen3 14B and 32B (Yang et al., 2025), Qwen2.5 70B, (Qwen Team, 2024), Llama-3.3 70B (Dubey et al., 2024), Prometheus-v2 7B (Kim et al., 2024b), and Selene-1 70B (Alexandru et al., 2025). We selected these models as the most capable judge models within our compute budget, and Selene-1 and Prometheus were selected as representatives of LLMs specifically fine-tuned for judging response quality. For latent probing, we only evaluate Qwen3 14B, Phi-4, and Prometheus v2 7B due to the increased cost of extracting embeddings.

## 4.3 RESULTS ON PAIRWISE AND SINGLE-RATING BENCHMARKS

Due to space constraints, we report results for Phi-4, Qwen3 14B, Prometheus, Llama 3.3 70B, and Selene-1, as well as only the 10-scale baseline. Also here, we report for models of varying sizes (from 7B to 70B parameters). Results for other models and the 5-scale baseline confirm observations and are in Appendix A.2.

**Pairwise Benchmarks.** Table 2 reports accuracies across pairwise evaluation benchmarks. Four key observations stand out: *First*, probability-weighted and verifier-style scores consistently match or exceed the 10-scale baseline, with gains of up to 5 percentage points in average accuracy. This indicates that extracting probability distributions or binary logits yields more discriminative signals than discrete categorical outputs, while remaining competitive where the baseline performs well. *Second*, specialized judge models such as Selene and Prometheus do not surpass general-purpose models (e.g., Llama, Phi-4). In fact, Prometheus fails entirely under probability-weighted and verifier setups, as it does not follow these prompts. This demonstrates that fine-tuning for judgment can compromise general model capabilities, reducing its applicability to alternative scoring schemes.

*Third*, when raw logits are poorly calibrated (e.g., Qwen3 14B under weighted scoring, Prometheus across settings), latent probes recover useful internal signals. By training directly on hidden activations, probes extract stable and fine-grained information about response quality that is not accessible through the model's output probabilities alone. This highlights latent probing as a general solution that can be applied to any judge model, regardless of its calibration. We also want to note that the latent probe can be further enhanced, to some degree (typically 1-2%) by targeted hyperparameter tuning, but here we report results for the same parameter configurations. *Fourth*, the comparison is conservative with respect to alternative methods: for the 10-scale baseline, frequent ties are resolved by random breaking. Thus, a model that produces ties in half of all cases already reaches 65% accuracy with only 40% correct and 10% incorrect predictions. The true discriminative ability of such baselines is therefore substantially lower than the reported numbers.

**Single-Rating Benchmarks** Table 3 reports Pearson correlations with ground-truth ratings, which themselves come from 5-point Likert scales or their averages. Here, we find that probability-weighted ratings and 10-scale ratings achieve the highest correlations, with strong models (Phi-4, Llama, Selene) reaching averages near 0.6. This suggests moderate to high correlation with human or GPT-4 ratings. However, verifier-style scores perform significantly worse. Verifyer-style scores mostly concentrate close to 0 or 1, leading to consistently lower correlations. For latent probes, the BCE objective used for training likely also squashes outputs toward the extremes, reducing variance

| Setting | | Benchmarks | | | | | | | | | | Average |
|---|---|---|---|---|---|---|---|---|---|---|---|---|
| | | Auto-J | HHH | JB | LFQA | MTB | PB | RB-2 | RB | Tülu | UF | |
| **Verifier** | Prometheus | 0.12 | 0.16 | 0.00 | 0.00 | 0.08 | 0.02 | 0.10 | 0.07 | 0.08 | 0.09 | 0.07 |
| | Qwen3 14B | 0.76 | 0.85 | 0.69 | 0.72 | 0.64 | 0.85 | **0.88** | 0.87 | 0.74 | 0.78 | 0.78 |
| | Phi-4 | 0.72 | 0.88 | 0.67 | 0.59 | 0.64 | 0.87 | 0.86 | 0.87 | 0.74 | 0.77 | 0.76 |
| | Llama 3.3 70B | 0.72 | 0.87 | 0.63 | 0.72 | 0.65 | 0.83 | 0.82 | 0.82 | 0.72 | 0.74 | 0.75 |
| | Selene-1 | 0.72 | 0.85 | 0.66 | 0.47 | 0.66 | 0.85 | 0.85 | 0.85 | 0.73 | 0.76 | 0.74 |
| **Weighted** | Prometheus | 0.00 | 0.00 | 0.00 | 0.00 | 0.00 | 0.00 | 0.00 | 0.00 | 0.00 | 0.00 | 0.00 |
| | Qwen3 14B | 0.66 | 0.83 | 0.43 | **0.77** | 0.57 | 0.84 | 0.58 | 0.59 | 0.54 | 0.61 | 0.64 |
| | Phi-4 | 0.77 | 0.89 | 0.69 | **0.77** | 0.67 | 0.92 | 0.85 | 0.89 | 0.75 | 0.82 | 0.80 |
| | Llama 3.3 70B | 0.79 | 0.91 | 0.65 | **0.77** | 0.67 | 0.91 | 0.85 | 0.89 | 0.75 | 0.83 | 0.80 |
| | Selene-1 70B | 0.79 | **0.92** | 0.68 | 0.76 | 0.68 | 0.94 | 0.86 | **0.91** | 0.77 | **0.85** | **0.82** |
| **10-Scale** | Prometheus | 0.67 | 0.71 | 0.54 | 0.64 | 0.60 | 0.89 | 0.51 | 0.71 | 0.62 | 0.63 | 0.66 |
| | Qwen3 14B | 0.77 | 0.78 | 0.61 | 0.74 | 0.65 | 0.81 | 0.73 | 0.82 | 0.73 | 0.79 | 0.74 |
| | Phi-4 | 0.74 | 0.83 | 0.59 | 0.76 | 0.65 | 0.84 | 0.73 | 0.83 | 0.72 | 0.76 | 0.74 |
| | Llama 3.3 70B | 0.77 | 0.85 | 0.60 | 0.75 | 0.65 | 0.88 | 0.72 | 0.84 | 0.73 | 0.78 | 0.76 |
| | Selene-1 | 0.77 | 0.87 | 0.63 | 0.74 | 0.65 | 0.91 | 0.76 | 0.88 | 0.75 | 0.82 | 0.78 |
| **Latent Probe** | Prometheus (Models) | 0.75 | 0.75 | 0.80 | 0.62 | 0.60 | 0.93 | 0.77 | 0.68 | 0.85 | 0.57 | 0.73 |
| | Prometheus (Tülu) | 0.74 | 0.75 | 0.79 | 0.61 | 0.62 | **0.95** | 0.77 | 0.69 | 0.83 | 0.57 | 0.73 |
| | Qwen3 14B (Models) | 0.75 | 0.70 | 0.89 | 0.64 | **0.87** | 0.89 | 0.79 | 0.76 | 0.87 | 0.69 | 0.78 |
| | Qwen3 14B (Tülu) | 0.75 | 0.72 | **0.90** | 0.66 | 0.86 | 0.88 | 0.78 | 0.74 | 0.86 | 0.68 | 0.78 |
| | Phi-4 (Models) | 0.78 | 0.75 | 0.88 | 0.66 | 0.85 | 0.90 | 0.80 | 0.76 | 0.88 | 0.67 | 0.79 |
| | Phi-4 (Tülu) | **0.80** | 0.76 | 0.88 | 0.67 | 0.84 | 0.91 | 0.82 | 0.76 | **0.89** | 0.68 | 0.80 |

Table 2: Pairwise evaluation accuracies on triplet datasets ($\langle$prompt, chosen, rejected$\rangle$): Auto-J, HHH-Alignment (HHH), JudgeBench (JB), LFQA, MT-Bench (MTB), PreferenceBench (PB), RewardBench-2 (RB-2), RewardBench (RB), Tülu Mixture (Tülu), and UltraFeedback (UF). Latent Probes require training; we denote the data source as Tülu = Preference pairs from (Lambert et al., 2024) and Models = generated by strong /weak LLMs. Best scores for each benchmark are in bold.

and degrading linear correlation. Combined with the discretization of ground truth, this limits their effectiveness on single-rating benchmarks.

**Summary.** Our extensive results on a large number of benchmarks show that approaches using internal judge representations serve as a valid replacement for traditional generative judges by addressing their limitations, such as non-determinism and saturated scales. Pairwise evaluation results show that their discriminativeness is superior or on par with the baselines. The linear correlation for single-rating evaluations is decent, especially for probability-weighted ratings. This is notable because the benchmarks closely resemble Likert scale ratings, which favors the 10-scale and 5-scale baselines. Finally, latent probing can recover model capabilities that are not accessible in training-free methods due to their miscalibrated outputs.

## 4.4 Ablations and Additional Analyses

**Are Scores Based on Internal Features Less Saturated?** In Section 3.1 (see Fig. 2), we observe that prompting judge models for integer ratings from 1 to 10 mostly produces scores near the top of the scale, such as between 7 and 9. In contrast, probability-weighted and verifier-style ratings are real-valued and thus more fine-grained. However, they are still bounded: probability-weighted scores by their scale (0–10 in our case) and verifier-style ratings between 0 and 1. As shown in Fig. 3, verifier-style scores are generally close to either 0 or 1. The variance of probability-weighted scores is small for Selene-1 and Llama, while it is significantly wider for the Qwen and Phi-4 models. Most importantly, these real-valued scores can be arbitrarily scaled and shifted without affecting their ordinal properties. This effectively solves the calibration issues observed with ordinal ratings.

**Calibration of Verifier-Style Scores.** Verifier-style scores concentrating near 0 or 1 degrades single-rating performance. To mitigate this, we apply temperature calibration (Guo et al., 2017) to the rating $S_b$ by taking the logits of "yes" ($z_{\text{yes}}$) and using the temperature-scaled sigmoid function $S_b = \sigma\left(\frac{z_{\text{yes}}}{T}\right)$. For each model-dataset pair, we optimize the temperature $T \in [1, 30]$ by L-BFGS-B to minimize the mean squared error between the predicted ratings and ground-truth scores on a validation split (5% of the data). As shown in Table 3, calibration consistently increases Pearson

| Setting | | Benchmarks | | | | | | | Average |
|---|---|---|---|---|---|---|---|---|---|
| | | BigGen-H | BigGen-J | Flask-G | Flask-H | MTB | UF | Vicuna | |
| **Verifier** | Prometheus | 0.00 | 0.00 | 0.06 | 0.07 | — | 0.03 | — | 0.03 |
| | *+ calibrated* | *0.00 (+0.00)* | *0.02 (+0.01)* | *0.12 (+0.06)* | *0.10 (+0.03)* | *—* | *0.04 (+0.01)* | *—* | *0.05 (+0.02)* |
| | Qwen3 14B | 0.43 | 0.64 | 0.37 | 0.37 | 0.32 | 0.63 | 0.44 | 0.46 |
| | *+ calibrated* | *0.47 (+0.05)* | *0.70 (+0.06)* | *0.42 (+0.05)* | *0.40 (+0.03)* | *0.55 (+0.23)* | *0.71 (+0.08)* | *0.50 (+0.06)* | *0.54 (+0.08)* |
| | Phi-4 | 0.46 | 0.70 | 0.42 | 0.37 | 0.54 | 0.66 | 0.50 | 0.52 |
| | *+ calibrated* | *0.49 (+0.03)* | *0.74 (+0.04)* | *0.46 (+0.04)* | *0.39 (+0.02)* | *0.70 (+0.16)* | *0.72 (+0.06)* | *0.60 (+0.10)* | *0.59 (+0.07)* |
| | Llama 3.3 70B | 0.39 | 0.66 | 0.44 | 0.35 | 0.56 | 0.70 | 0.60 | 0.53 |
| | *+ calibrated* | *0.43 (+0.04)* | *0.70 (+0.04)* | *0.43 (-0.01)* | *0.35 (-0.00)* | *0.64 (+0.08)* | *0.76 (+0.06)* | *0.59 (-0.01)* | *0.56 (+0.03)* |
| | Selene-1 | 0.42 | 0.68 | 0.45 | 0.36 | 0.56 | 0.70 | 0.48 | 0.52 |
| | *+ calibrated* | *0.46 (+0.04)* | *0.72 (+0.05)* | *0.46 (+0.02)* | *0.36 (-0.01)* | *0.67 (+0.12)* | *0.77 (+0.07)* | *0.58 (+0.10)* | *0.57 (+0.05)* |
| **Weighted** | Prometheus | — | — | — | — | — | — | — | — |
| | Qwen3 14B | −0.06 | 0.18 | 0.27 | 0.29 | 0.53 | 0.55 | 0.24 | 0.29 |
| | Phi-4 | 0.47 | 0.73 | 0.51 | 0.50 | 0.72 | 0.75 | 0.49 | 0.59 |
| | Llama 3.3 70B | 0.43 | 0.69 | 0.46 | 0.46 | 0.83 | 0.75 | 0.59 | 0.60 |
| | Selene-1 70B | 0.45 | 0.71 | **0.54** | **0.54** | **0.87** | 0.79 | 0.60 | **0.64** |
| **10-Scale** | Prometheus | 0.31 | 0.53 | 0.21 | 0.24 | 0.50 | — | 0.39 | 0.36 |
| | Qwen3 14B | 0.47 | 0.69 | 0.43 | 0.45 | 0.69 | 0.72 | 0.28 | 0.53 |
| | Phi-4 | 0.44 | 0.72 | 0.46 | **0.54** | 0.76 | 0.74 | 0.44 | 0.59 |
| | Llama 3.3 70B | 0.44 | 0.70 | 0.53 | 0.51 | 0.75 | 0.76 | **0.61** | 0.61 |
| | Selene-1 | 0.47 | 0.71 | 0.51 | 0.51 | 0.78 | **0.81** | 0.60 | 0.63 |
| **Latent Probe** | Prometheus (Models) | 0.36 | 0.60 | 0.20 | 0.24 | 0.52 | 0.60 | 0.21 | 0.39 |
| | Prometheus (Tülu) | 0.35 | 0.60 | 0.23 | 0.26 | 0.53 | 0.64 | 0.23 | 0.40 |
| | Qwen3 14B (Models) | 0.46 | 0.70 | 0.32 | 0.35 | 0.55 | 0.73 | 0.23 | 0.48 |
| | Qwen3 14B (Tülu) | 0.45 | 0.71 | 0.31 | 0.36 | 0.55 | 0.74 | 0.23 | 0.48 |
| | Phi-4 (Models) | **0.49** | 0.73 | 0.31 | 0.32 | 0.46 | 0.76 | 0.21 | 0.47 |
| | Phi-4 (Tülu) | **0.49** | **0.74** | 0.33 | 0.33 | 0.43 | 0.74 | 0.15 | 0.46 |

Table 3: Single-response rating correlations with ground-truth scores on BigGen (human: BigGen-H, GPT: BigGen-J), FLASK (GPT: Flask-G, human: Flask-H), MT-Bench (MTB), UltraFeedback (UF), and Vicuna-Eval (Vicuna). Entries are Pearson correlations between model-produced scores and reference ratings. Calibrated rows apply temperature scaling to verifier scores. Latent Probes require training; we denote the data source as Tülu = Preference pairs from (Lambert et al., 2024) and Models = generated by strong /weak LLMs. Best scores for each benchmark are in bold.

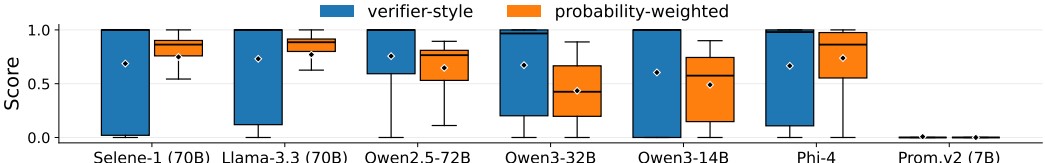

Figure 3: Calibration behavior across models for verifier-style and probability-weighted ratings. Probability-weighted ratings are rescaled between 0 and 10, while their original values are between 0 and 10.

correlations across all judge models. For example, Qwen3-14B improved from 0.46 to 0.54 on average. Crucially, the calibrated verifier-style scores now nearly match the performance of probability-weighted scores (e.g., Phi-4 achieves 0.59 in both settings), highlighting that our methods are open to further improvement in future work.

**Data Requirements of Latent Probes.** In Table 2 and Table 3, we present two types of latent probes. The first is trained on embeddings from the Tülu Preference Mixture (Lambert et al., 2024), which consists of preferred and rejected responses derived from costly, GPT-4-annotated rubrics. The second is trained on unsupervised data, where positive examples (preferred responses) are generated by strong LLMs, such as ExaOne-3.5 7.8B and GLM-4 9B, and negative examples (rejected responses) are generated by weak LLMs, such as Llama 3.2 3B and Qwen3 4B. The choice of the model pair for generating unsupervised preference pairs is important, as more clearly separated models (in terms of performance) make training more stable. We notice that probe performance can vary depending on the chosen data and hyperparameters. Fortunately, validated preference pairs are now widely available (Cui et al., 2024; Lambert et al., 2024), and we can construct model pairs with a sufficiently large performance gap (Geng et al., 2025).

|  | Verifier | Weighted | 10-Scale | 5-Scale | Latent Probe | |
|---|---|---|---|---|---|---|
|  |  |  |  |  | Models | Tülu |
| Prometheus | -0.045 | — | 0.25 | 0.25 | **0.38** | 0.36 |
| Qwen3 14B | 0.41 | -0.15 | 0.40 | 0.36 | 0.37 | **0.46** |
| Phi-4 | 0.39 | 0.42 | 0.38 | 0.35 | 0.33 | **0.43** |
| Llama-3.3 70B | 0.31 | **0.45** | 0.30 | 0.30 | — | — |
| Selene-1 | 0.35 | **0.48** | 0.40 | 0.42 | — | — |

Table 4: Spearman rank correlation ($\rho$) between rankings from rating methods and GPT-5 reference rankings on 1000 prompts from the Tülu Preference Mixture dataset. Higher values indicate closer agreement with GPT-5 rankings.

## 5 APPLICATIONS

In this section, we provide evidence that LLM judges operating on latent information improve downstream tasks. In Section 5.1, we show that probability-weighted scores and latent probes outperform the traditional paradigm on listwise ranking, i.e., the ranking of multiple responses from different LLMs. In Section 5.2, we demonstrate that prompt semantics contain only a limited signal for predicting model performance in free-generation tasks. This extends routing beyond the typical discriminative setting, where we can objectively measure model performance (Zhang et al., 2025b).

### 5.1 LISTWISE RANKING

Methods such as Best-of-$N$, multi-teacher distillation, and GRPO require ranking or scoring responses, for example, to select the best one. We, therefore, evaluate how well ratings assigned by various methods agree with GPT-5 rankings. Specifically, we generate responses to 1000 prompts from the Tülu Preference Mixture dataset (Lambert et al., 2024) using 22 LLMs. We then obtain a ranking of these responses (a total ordering without ties) for each prompt from GPT-5-Mini with high reasoning effort. All methods assign a score to each response, and we compare the resulting rankings to the GPT-5 rankings using Spearman's rank correlation ($\rho$).

Results in Table 4 show that methods based on latent information, especially probability-weighted ratings and latent probes, clearly outperform the baseline of ordinal ratings. However, as observed previously, Qwen3-14 does not yield useful probability-weighted ratings. Furthermore, latent probes trained with Tülu responses are significantly more effective than probes trained on other generated responses. This may be due to in-distribution effects, as both training and evaluation prompts come from the Tülu dataset, although we keep training and test prompts separate. Overall, these results underscore advantages of latent information methods over traditional ordinal ratings.

### 5.2 LLM ROUTING

Given an input query, routing attempts to select the most suitable LLM from a pool of models to generate an answer (Ong et al., 2025). The primary applications of routing are to optimize cost-performance tradeoffs (Stripelis et al., 2024; Kassem et al., 2025; Wang et al., 2025; Panda et al., 2025) or to improve performance by leveraging the complementary strengths of different LLMs (Ong et al., 2025; Zhang et al., 2025b). Combined with ensembling, Zhang et al. (2025b) demonstrates strong routing performance on knowledge-focused benchmarks. However, previous work has generally focused on queries with objectively correct answers, such as those in math, coding, or factual knowledge. More complex aspects like helpfulness, informativeness, and prompt following have received less attention. Therefore, we evaluate the feasibility of learning simple routers for general response quality beyond verifiable correctness. We generate responses to 200K prompts from the UltraChat dataset (Ding et al., 2023) using 16 small LLMs ranging from 3B to 14B. We then score each response using Phi-4. Next, we embed all UltraChat prompts using `ibm-granite/granite-embedding-english-r2` (Awasthy et al., 2025). We then calculate how well the weighted average score of the 50 nearest neighbor prompts in the embedding space predicts the score of a given prompt.

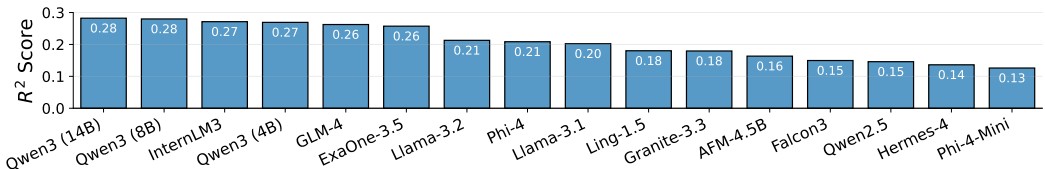

Figure 4: $R^2$ scores measuring how well the weighted average response quality of the 50 nearest-neighbor prompts in embedding space predicts the response quality of a given prompt across LLMs.

The results ($R^2$ scores) in Fig. 4 indicate that while the semantic similarity of prompts is somewhat predictive of model performance for all models, the correlation is not very strong. We confirm this by training $k$-nn routers (Li et al., 2025) to predict which model should generate the answer, and we find that prompt embeddings alone are not informative enough to raise performance above that of the best individual model. This motivates research into more advanced routing mechanisms that use more information than just prompt semantics.

## 6 DISCUSSION AND LIMITATIONS

Our analysis demonstrates that LLM judges operating on latent information outperform traditional Likert-style metrics. Empirical results in Section 4.3 and Section 5.1 support this claim. These findings suggest that LLM evaluation and applications benefit from using internal activations, such as lightweight probes or logits, rather than generated output. However, the optimal method is task-dependent. For instance, in Table 2, Table 3, and Table 4, we find that probability-weighted scores or latent probes generally perform best. This suggests the potential to expand our research by examining the specific mechanisms through which latent information enhances evaluation performance and how to capitalize on it for improved results.

One important question is *why* latent information provides a more discriminative signal for evaluating responses. We argue that it is inherently more fine-grained, as sampling from the judge LLM to obtain ratings loses information. This process simplifies the representation of rich internal responses. Logits and activations encode richer features that distinguish between responses. These include stylistic, semantic, or discursive features. More direct access to these attributes facilitates more reliable response assessment.

Finally, regarding computational costs, verifier-style ratings and probability-weighted scores incur no overhead over standard inference. While probe training requires initial inference for data preparation, training linear probes on cached activations is negligible. Moreover, latent probes reduce inference costs. For example, applying a linear probe at layer 31 of 40 (as with Phi-4) allows us to skip subsequent layers, saving approximately 25% of the forward pass.

## 7 CONCLUSION

In this paper, we systematically examine the weaknesses of traditional ordinal ratings assigned by an LLM-as-a-judge. We find that these ratings can be unstable under sampling and tend to use only a small range of the available scores. This makes them less practical in settings that require unambiguous, reference-free response ratings. Such settings are highly relevant, as they include methods like Best-of-$N$ sampling, GRPO, and multi-teacher distillation, which enable significant improvements in important applications.

We then show that latent judges, whose ratings are based on model logits or internal activations, perform as well as or better than the traditional LLM-as-a-judge approach. We validate this across a wide range of pairwise and single-rating benchmarks. We also extend our evaluation to the practically relevant listwise rating and demonstrate how latent judges can be used in LLM routers.

Our insights are relevant and can inform stronger methods for the applications listed. Future research can investigate specific fine-tuning methods for latent judges, their robustness to known weaknesses of reward models such as reward hacking, and their downstream applicability in these applications.

## REPRODUCIBILITY STATEMENT

Code and data to reproduce our experiments and findings will be publicly released on acceptance. Prompts used in our experiments are documented in Appendix C. To improve clarity, the manuscript was polished for grammar and style using a large language model, with all final text reviewed and validated by the authors.

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

# Supplementary Material

## A ADDITIONAL RESULTS

### A.1 ABLATING THE IMPACT OF TEMPERATURE ON SCORE STABILITY

In Section 3.1, we demonstrate that LLM ratings are unstable under stochastic sampling. However, this instability depends on the generation temperature: higher temperatures produce more variable outputs, while a temperature of 0 yields deterministic outputs (subject to low-level library or hardware non-determinism). Here, we evaluate how the specific temperature choice affects this instability. We repeat the evaluation from Section 3.1 using three models (Llama-3.3 70B, Selene-1 70B, and Phi-4) across different temperatures $\tau \in \{0.1, 0.2, 0.4, 0.6, 0.8, 1.0\}$.

Fig. 5 displays the resulting agreement with the mode rating. As expected, instability decreases at lower temperatures. However, even at $\tau = 0.1$, responses from Llama-3.3 70B and Phi-4 disagree with the mode rating approximately one out of five times. Selene-1 retains the relatively greater stability observed in Fig. 1. In no instance is the stability truly deterministic, which confirms our observation of significant excess epistemic uncertainty in all judge models.

In Table 5, we show that agreement with GPT-4 judgments on the Tülu Preference dataset remains largely unaffected by temperature. Llama-3.3 70B and Selene 70B show essentially no variation across temperatures, while Phi-4 displays limited variability (the difference between $\tau = 0.1$ and $\tau = 1.0$ ranges from 3% to 4%). Strict accuracy (ties not allowed) and lenient accuracy (ties counted as correct) exhibit opposing trends: strict accuracy increases with higher temperatures, whereas lenient accuracy decreases. This occurs because higher diversity reduces the frequency of ties. In lenient evaluation, fewer ties result in fewer "correct" outcomes. Conversely, in strict evaluation, reduced ties and increased diversity raise the probability of identifying the correct ordering.

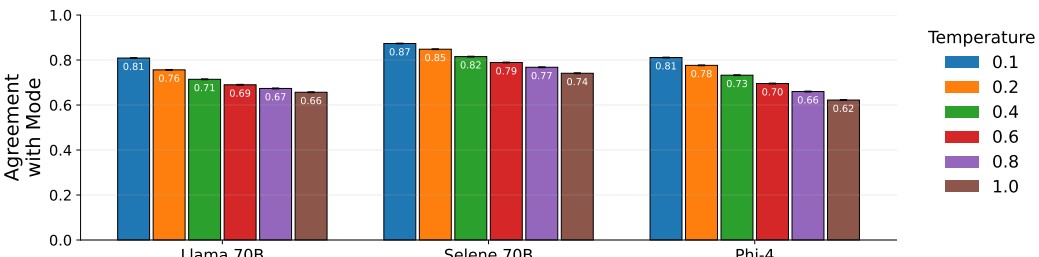

Figure 5: Agreement with mode rating across sampling temperatures. We report the stability of model judgments, defined as the proportion of generated ratings that match the mode rating, for Llama-3.3 70B, Selene-1 70B, and Phi-4 across temperatures $\tau \in \{0.1, \ldots, 1.0\}$.

| | **Strict** | | | | | | **Lenient** | | | | | |
| Model | 0.1 | 0.2 | 0.4 | 0.6 | 0.8 | 1.0 | 0.1 | 0.2 | 0.4 | 0.6 | 0.8 | 1.0 |
|---|---|---|---|---|---|---|---|---|---|---|---|---|
| Llama 70B | 60.9 | 61.1 | 60.9 | 60.8 | 61.2 | 61.3 | 88.5 | 88.5 | 88.3 | 88.1 | 88.0 | 87.8 |
| Phi-4 | 38.9 | 39.6 | 41.0 | 42.1 | 42.7 | 43.4 | 84.9 | 84.4 | 83.6 | 82.9 | 81.9 | 81.4 |
| Selene 70B | 57.0 | 57.2 | 56.8 | 56.6 | 56.3 | 56.3 | 91.2 | 90.9 | 90.7 | 90.4 | 90.2 | 89.7 |

Table 5: Agreement with GPT-4 judgments across sampling temperatures. We report strict and lenient accuracy scores on the Tülu Preference dataset for temperatures $\tau \in \{0.1, \ldots, 1.0\}$. Strict accuracy counts ties as incorrect, whereas lenient accuracy treats ties as matches.

### A.2 FULL RESULTS FOR PAIRWISE AND SINGLE-RATING BENCHMARKS

Full results, including all models and 5-scale baselines, on pairwise and single-rating benchmarks are in Table 6 (pairwise) and in Table 7 (single-rating).

| Setting | | Benchmarks | | | | | | | | | | Average |
|---|---|---|---|---|---|---|---|---|---|---|---|---|
| | | Auto-J | HHH | JB | LFQA | MTB | PB | RB-2 | RB | Tülu | UF | |
| Verifier | Prometheus-7B v2.0 | 0.12 | 0.16 | 0.00 | 0.00 | 0.08 | 0.02 | 0.10 | 0.07 | 0.08 | 0.09 | 0.07 |
| | Phi-4 | 0.72 | 0.88 | 0.67 | 0.59 | 0.64 | 0.87 | 0.86 | 0.87 | 0.74 | 0.77 | 0.76 |
| | Qwen3 14B | 0.76 | 0.85 | 0.69 | 0.72 | 0.64 | 0.85 | 0.88 | 0.87 | 0.74 | 0.78 | 0.78 |
| | Qwen3 32B | 0.74 | 0.92 | 0.75 | 0.77 | 0.66 | 0.88 | 0.91 | 0.89 | 0.75 | 0.79 | 0.80 |
| | Qwen2.5 72B | 0.74 | 0.89 | 0.71 | 0.72 | 0.66 | 0.87 | 0.90 | 0.90 | 0.76 | 0.80 | 0.80 |
| | Llama-3.3 70B | 0.72 | 0.87 | 0.63 | 0.72 | 0.65 | 0.83 | 0.82 | 0.82 | 0.72 | 0.74 | 0.75 |
| | Selene-1 70B | 0.72 | 0.85 | 0.66 | 0.47 | 0.66 | 0.85 | 0.85 | 0.85 | 0.73 | 0.76 | 0.74 |
| Weighted | Prometheus-7B v2.0 | 0.00 | 0.00 | 0.00 | 0.00 | 0.00 | 0.00 | 0.00 | 0.00 | 0.00 | 0.00 | 0.00 |
| | Phi-4 | 0.77 | 0.89 | 0.69 | 0.77 | 0.67 | 0.92 | 0.85 | 0.89 | 0.75 | 0.82 | 0.80 |
| | Qwen3 14B | 0.66 | 0.83 | 0.43 | 0.77 | 0.57 | 0.84 | 0.58 | 0.59 | 0.54 | 0.61 | 0.64 |
| | Qwen3 32B | 0.54 | 0.83 | 0.39 | 0.78 | 0.54 | 0.76 | 0.47 | 0.48 | 0.48 | 0.48 | 0.57 |
| | Qwen2.5 72B | 0.74 | 0.89 | 0.45 | 0.78 | 0.63 | 0.90 | 0.68 | 0.73 | 0.65 | 0.70 | 0.71 |
| | Llama-3.3 70B | 0.79 | 0.91 | 0.65 | 0.77 | 0.67 | 0.91 | 0.85 | 0.89 | 0.75 | 0.83 | 0.80 |
| | Selene-1 70B | 0.79 | 0.92 | 0.68 | 0.76 | 0.68 | 0.94 | 0.86 | 0.91 | 0.77 | 0.85 | 0.82 |
| 10-Scale | Prometheus-7B v2.0 | 0.67 | 0.71 | 0.54 | 0.64 | 0.60 | 0.89 | 0.51 | 0.71 | 0.62 | — | 0.66 |
| | Phi-4 | 0.74 | 0.83 | 0.59 | 0.76 | 0.65 | 0.84 | 0.73 | 0.83 | 0.72 | 0.76 | 0.74 |
| | Qwen3 14B | 0.77 | 0.78 | 0.61 | 0.74 | 0.65 | 0.81 | 0.73 | 0.82 | 0.73 | 0.79 | 0.74 |
| | Qwen3 32B | 0.77 | 0.82 | 0.59 | 0.75 | 0.63 | 0.85 | 0.76 | 0.82 | 0.73 | 0.79 | 0.75 |
| | Qwen2.5 72B | 0.76 | 0.86 | 0.60 | 0.76 | 0.66 | 0.86 | 0.71 | 0.85 | 0.73 | 0.79 | 0.76 |
| | Llama-3.3 70B | 0.77 | 0.85 | 0.60 | 0.75 | 0.65 | 0.88 | 0.72 | 0.84 | 0.73 | 0.78 | 0.76 |
| | Selene-1 70B | 0.77 | 0.87 | 0.63 | 0.74 | 0.65 | 0.91 | 0.76 | 0.88 | 0.75 | 0.82 | 0.78 |
| 5-Scale | Prometheus-7B v2.0 | 0.65 | 0.67 | 0.71 | 0.57 | 0.52 | 0.87 | – | 0.62 | 0.69 | 0.55 | 0.65 |
| | Phi-4 | 0.69 | 0.75 | 0.81 | 0.64 | 0.73 | 0.79 | 0.70 | 0.73 | 0.77 | 0.62 | 0.72 |
| | Qwen3 14B | 0.69 | 0.72 | 0.82 | 0.64 | 0.74 | 0.79 | 0.74 | 0.71 | 0.79 | 0.61 | 0.72 |
| | Qwen3 32B | 0.72 | 0.74 | 0.80 | 0.63 | 0.74 | 0.79 | 0.74 | 0.70 | 0.78 | 0.61 | 0.72 |
| | Qwen2.5 72B | 0.73 | 0.75 | 0.83 | 0.64 | 0.72 | 0.80 | 0.73 | 0.71 | 0.79 | 0.60 | 0.73 |
| | Llama-3.3 70B | 0.71 | 0.74 | 0.82 | 0.64 | 0.72 | 0.81 | 0.75 | 0.71 | 0.81 | 0.62 | 0.73 |
| | Selene-1 70B | 0.74 | 0.73 | 0.82 | 0.65 | 0.76 | 0.84 | 0.77 | 0.73 | 0.85 | 0.63 | 0.75 |
| Latent Probe | Prometheus (Models) | 0.75 | 0.75 | 0.80 | 0.62 | 0.60 | 0.93 | 0.77 | 0.68 | 0.85 | 0.57 | 0.73 |
| | Prometheus (Tülu) | 0.74 | 0.75 | 0.79 | 0.61 | 0.62 | 0.95 | 0.77 | 0.69 | 0.83 | 0.57 | 0.73 |
| | Qwen3 14B (Models) | 0.75 | 0.70 | 0.89 | 0.64 | 0.87 | 0.89 | 0.79 | 0.76 | 0.87 | 0.69 | 0.78 |
| | Qwen3 14B (Tülu) | 0.75 | 0.72 | 0.90 | 0.66 | 0.86 | 0.88 | 0.78 | 0.74 | 0.86 | 0.68 | 0.78 |
| | Phi-4 (Models) | 0.78 | 0.75 | 0.88 | 0.66 | 0.85 | 0.90 | 0.80 | 0.76 | 0.88 | 0.67 | 0.79 |
| | Phi-4 (Tülu) | 0.80 | 0.76 | 0.88 | 0.67 | 0.84 | 0.91 | 0.82 | 0.76 | 0.89 | 0.68 | 0.80 |

Table 6: Full pairwise evaluation accuracies on triplet datasets (⟨prompt, chosen, rejected⟩): Auto-J, HHH-Alignment (HHH), JudgeBench (JB), LFQA, MT-Bench (MTB), PreferenceBench (PB), RewardBench-2 (RB-2), RewardBench (RB), Tülu Mixture (Tülu), and UltraFeedback (UF).

| Setting | | Benchmarks | | | | | | | Average |
|---|---|---|---|---|---|---|---|---|---|
| | | BigGen-H | BigGen-J | Flask-G | Flask-H | MTB | UF | Vicuna | |
| Verifier-Style | Prometheus | 0.00 | 0.00 | 0.06 | 0.07 | — | 0.03 | — | 0.03 |
| | Phi-4 | 0.46 | 0.70 | 0.42 | 0.37 | 0.54 | 0.66 | 0.50 | 0.52 |
| | Qwen3 14B | 0.43 | 0.64 | 0.37 | 0.37 | 0.32 | 0.63 | 0.44 | 0.46 |
| | Qwen3-32B | 0.48 | 0.69 | 0.42 | 0.38 | 0.47 | 0.71 | 0.51 | 0.52 |
| | Qwen2.5 72B | 0.43 | 0.69 | 0.52 | 0.53 | 0.78 | 0.75 | 0.55 | 0.61 |
| | Llama 3.3 70B | 0.39 | 0.66 | 0.44 | 0.35 | 0.56 | 0.70 | 0.60 | 0.53 |
| | Selene-1 | 0.42 | 0.68 | 0.45 | 0.36 | 0.56 | 0.70 | 0.48 | 0.52 |
| Weighted | Prometheus | — | — | — | — | — | — | — | — |
| | Phi-4 | 0.47 | 0.73 | 0.51 | 0.50 | 0.72 | 0.75 | 0.49 | 0.59 |
| | Qwen3 14B | −0.06 | 0.18 | 0.27 | 0.29 | 0.53 | 0.55 | 0.24 | 0.29 |
| | Qwen3 32B | −0.11 | 0.10 | 0.06 | 0.09 | 0.62 | 0.47 | −0.10 | 0.16 |
| | Qwen2.5 72B | 0.21 | 0.53 | 0.52 | 0.54 | 0.81 | 0.74 | 0.30 | 0.52 |
| | Llama 3.3 70B | 0.43 | 0.69 | 0.46 | 0.46 | 0.83 | 0.75 | 0.59 | 0.60 |
| | Selene-1 70B | 0.45 | 0.71 | 0.54 | 0.54 | 0.87 | 0.79 | 0.60 | 0.64 |
| 10-Scale | Prometheus | 0.31 | 0.53 | 0.21 | 0.24 | 0.50 | — | 0.39 | 0.36 |
| | Phi-4 | 0.44 | 0.72 | 0.46 | 0.54 | 0.76 | 0.74 | 0.44 | 0.59 |
| | Qwen3 14B | 0.47 | 0.69 | 0.43 | 0.45 | 0.69 | 0.72 | 0.28 | 0.53 |
| | Qwen3 32B | 0.45 | 0.71 | 0.43 | 0.51 | 0.70 | 0.70 | 0.38 | 0.55 |
| | Qwen2.5 72B | 0.44 | 0.70 | 0.52 | 0.56 | 0.86 | 0.78 | 0.51 | 0.62 |
| | Llama 3.3 70B | 0.44 | 0.70 | 0.53 | 0.51 | 0.75 | 0.76 | 0.61 | 0.61 |
| | Selene-1 | 0.47 | 0.71 | 0.51 | 0.51 | 0.78 | 0.81 | 0.60 | 0.63 |
| 5-Scale | Prometheus | 0.29 | — | 0.18 | 0.19 | 0.56 | 0.54 | 0.38 | 0.36 |
| | Phi-4 | 0.40 | 0.71 | 0.48 | 0.55 | 0.71 | 0.72 | 0.46 | 0.58 |
| | Qwen3 14B | 0.43 | 0.68 | 0.46 | 0.48 | 0.70 | 0.72 | 0.36 | 0.55 |
| | Qwen3 32B | 0.43 | 0.70 | 0.44 | 0.53 | 0.75 | — | 0.44 | 0.55 |
| | Qwen2.5 72B | 0.39 | 0.69 | 0.50 | 0.52 | 0.85 | 0.75 | 0.45 | 0.59 |
| | Llama 3.3 70B | 0.41 | 0.69 | 0.43 | 0.41 | 0.78 | 0.75 | 0.54 | 0.57 |
| | Selene-1 | 0.46 | 0.72 | 0.47 | 0.52 | 0.79 | 0.79 | 0.60 | 0.62 |
| Latent Probe | Prometheus (Models) | 0.36 | 0.60 | 0.20 | 0.24 | 0.52 | 0.60 | 0.21 | 0.39 |
| | Prometheus (Tülu) | 0.35 | 0.60 | 0.23 | 0.26 | 0.53 | 0.64 | 0.23 | 0.40 |
| | Qwen3 14B (Models) | 0.46 | 0.70 | 0.32 | 0.35 | 0.55 | 0.73 | 0.23 | 0.48 |
| | Qwen3 14B (Tülu) | 0.45 | 0.71 | 0.31 | 0.36 | 0.55 | 0.74 | 0.23 | 0.48 |
| | Phi-4 (Models) | 0.49 | 0.73 | 0.31 | 0.32 | 0.46 | 0.76 | 0.21 | 0.47 |
| | Phi-4 (Tülu) | 0.49 | 0.74 | 0.33 | 0.33 | 0.43 | 0.74 | 0.15 | 0.46 |

Table 7: Full single-response rating correlations with ground-truth scores on BigGen (human: BigGen-H, GPT: BigGen-J), FLASK (GPT: Flask-G, human: Flask-H), MT-Bench (MTB), UltraFeedback (UF), and Vicuna-Eval (Vicuna). Entries are Pearson correlations between model-produced scores and reference ratings.

## A.3 ADDITIONAL RESULTS FOR PAIRWISE BECHMARKS

**Confidence Intervals for Accuracies.** In Table 8, we add confidence intervals for benchmark accuracies in Table 2. Confidence intervals are computed as the 95% Clopper-Pearson exact interval, because the accuracy is equivalent to the success rate of a Bernoulli distribution.

| | | Auto-J | HHH | JB | LFQA | MTB | PB | RB | RB-2 | Tülu | UF |
|---|---|---|---|---|---|---|---|---|---|---|---|
| Verifier | Llama 3.3 70B | $0.72_{0.69-0.75}$ | $0.87_{0.82-0.91}$ | $0.63_{0.59-0.66}$ | $0.72_{0.69-0.75}$ | $0.65_{0.62-0.68}$ | $0.83_{0.81-0.84}$ | $0.82_{0.81-0.84}$ | $0.82_{0.80-0.84}$ | $0.72_{0.72-0.73}$ | $0.74_{0.73-0.75}$ |
| | Qwen3 14B | $0.76_{0.73-0.78}$ | $0.85_{0.79-0.89}$ | $0.69_{0.66-0.73}$ | $0.72_{0.69-0.75}$ | $0.64_{0.61-0.68}$ | $0.85_{0.83-0.86}$ | $0.87_{0.86-0.88}$ | $0.88_{0.86-0.89}$ | $0.74_{0.73-0.74}$ | $0.78_{0.78-0.79}$ |
| | Selene-1 | $0.72_{0.69-0.74}$ | $0.85_{0.80-0.89}$ | $0.66_{0.62-0.69}$ | $0.47_{0.44-0.50}$ | $0.66_{0.63-0.69}$ | $0.85_{0.83-0.87}$ | $0.87_{0.85-0.88}$ | $0.85_{0.84-0.87}$ | $0.73_{0.72-0.74}$ | $0.76_{0.75-0.77}$ |
| | Phi-4 | $0.72_{0.69-0.75}$ | $0.88_{0.83-0.92}$ | $0.67_{0.64-0.71}$ | $0.59_{0.56-0.62}$ | $0.64_{0.61-0.68}$ | $0.87_{0.85-0.88}$ | $0.87_{0.85-0.88}$ | $0.86_{0.85-0.88}$ | $0.74_{0.73-0.75}$ | $0.77_{0.76-0.77}$ |
| | Prometheus | $0.12_{0.10-0.15}$ | $0.16_{0.12-0.22}$ | $0.00_{0.00-0.00}$ | $0.00_{0.00-0.01}$ | $0.08_{0.07-0.10}$ | $0.02_{0.01-0.03}$ | $0.07_{0.07-0.08}$ | $0.10_{0.09-0.11}$ | $0.08_{0.08-0.09}$ | $0.09_{0.09-0.10}$ |
| Weighted | Llama 3.3 70B | $0.79_{0.77-0.82}$ | $0.91_{0.86-0.94}$ | $0.65_{0.62-0.69}$ | $0.77_{0.75-0.80}$ | $0.67_{0.63-0.70}$ | $0.91_{0.90-0.93}$ | $0.89_{0.88-0.90}$ | $0.85_{0.84-0.87}$ | $0.75_{0.75-0.76}$ | $0.83_{0.82-0.84}$ |
| | Qwen3 14B | $0.66_{0.63-0.69}$ | $0.83_{0.78-0.88}$ | $0.43_{0.39-0.47}$ | $0.77_{0.75-0.80}$ | $0.57_{0.54-0.61}$ | $0.84_{0.83-0.86}$ | $0.59_{0.58-0.61}$ | $0.58_{0.56-0.61}$ | $0.54_{0.53-0.55}$ | $0.61_{0.60-0.62}$ |
| | Selene-1 | $0.79_{0.77-0.82}$ | $0.92_{0.87-0.95}$ | $0.68_{0.64-0.72}$ | $0.76_{0.73-0.78}$ | $0.68_{0.65-0.71}$ | $0.94_{0.93-0.95}$ | $0.91_{0.90-0.92}$ | $0.86_{0.84-0.87}$ | $0.77_{0.76-0.78}$ | $0.85_{0.84-0.85}$ |
| | Phi-4 | $0.77_{0.75-0.80}$ | $0.89_{0.84-0.93}$ | $0.69_{0.66-0.73}$ | $0.77_{0.75-0.80}$ | $0.67_{0.64-0.70}$ | $0.92_{0.90-0.93}$ | $0.89_{0.87-0.90}$ | $0.85_{0.83-0.86}$ | $0.75_{0.75-0.76}$ | $0.82_{0.81-0.83}$ |
| | Prometheus | $0.00_{0.00-0.00}$ | $0.00_{0.00-0.00}$ | $0.00_{0.00-0.00}$ | $0.00_{0.00-0.01}$ | $0.00_{0.00-0.00}$ | $0.00_{0.00-0.00}$ | $0.00_{0.00-0.00}$ | $0.00_{0.00-0.00}$ | $0.00_{0.00-0.00}$ | $0.00_{0.00-0.00}$ |
| 10-Scale | Llama 3.3 70B | $0.77_{0.74-0.79}$ | $0.84_{0.79-0.89}$ | $0.60_{0.56-0.64}$ | $0.74_{0.72-0.77}$ | $0.65_{0.61-0.68}$ | $0.88_{0.87-0.89}$ | $0.84_{0.83-0.85}$ | $0.71_{0.69-0.74}$ | $0.72_{0.72-0.73}$ | $0.79_{0.78-0.79}$ |
| | Qwen3 14B | $0.76_{0.73-0.79}$ | $0.81_{0.75-0.86}$ | $0.62_{0.58-0.66}$ | $0.74_{0.72-0.77}$ | $0.64_{0.61-0.67}$ | $0.81_{0.79-0.83}$ | $0.81_{0.79-0.82}$ | $0.75_{0.73-0.77}$ | $0.73_{0.72-0.74}$ | $0.79_{0.79-0.80}$ |
| | Selene-1 | $0.76_{0.73-0.78}$ | $0.85_{0.79-0.89}$ | $0.60_{0.56-0.64}$ | $0.74_{0.71-0.77}$ | $0.65_{0.62-0.68}$ | $0.91_{0.90-0.93}$ | $0.88_{0.86-0.89}$ | $0.75_{0.73-0.77}$ | $0.75_{0.75-0.76}$ | $0.82_{0.81-0.83}$ |
| | Phi-4 | $0.72_{0.69-0.75}$ | $0.82_{0.77-0.87}$ | $0.57_{0.53-0.61}$ | $0.76_{0.74-0.79}$ | $0.65_{0.62-0.68}$ | $0.84_{0.82-0.86}$ | $0.83_{0.81-0.84}$ | $0.71_{0.69-0.74}$ | $0.71_{0.70-0.72}$ | $0.76_{0.75-0.77}$ |
| | Prometheus | $0.67_{0.64-0.70}$ | $0.73_{0.66-0.79}$ | $0.54_{0.50-0.58}$ | $0.64_{0.61-0.67}$ | $0.59_{0.56-0.62}$ | $0.89_{0.87-0.90}$ | $0.71_{0.70-0.73}$ | $0.52_{0.50-0.54}$ | $0.63_{0.62-0.63}$ | $0.62_{0.61-0.62}$ |
| Latent Probe | Qwen3 14B (Models) | $0.75_{0.72-0.78}$ | $0.89_{0.84-0.93}$ | $0.68_{0.64-0.72}$ | $0.73_{0.70-0.76}$ | $0.64_{0.61-0.67}$ | $0.89_{0.87-0.90}$ | $0.86_{0.84-0.87}$ | $0.86_{0.85-0.88}$ | $0.75_{0.74-0.76}$ | $0.78_{0.78-0.79}$ |
| | Qwen3 14B (Tülu) | $0.76_{0.73-0.78}$ | $0.88_{0.83-0.92}$ | $0.69_{0.65-0.72}$ | $0.68_{0.65-0.71}$ | $0.65_{0.61-0.67}$ | $0.89_{0.87-0.90}$ | $0.87_{0.86-0.88}$ | $0.87_{0.85-0.88}$ | $0.75_{0.74-0.76}$ | $0.79_{0.78-0.80}$ |
| | Phi-4 (Models) | $0.76_{0.73-0.78}$ | $0.90_{0.85-0.94}$ | $0.71_{0.67-0.75}$ | $0.75_{0.72-0.77}$ | $0.66_{0.63-0.69}$ | $0.91_{0.90-0.92}$ | $0.87_{0.86-0.88}$ | $0.83_{0.81-0.85}$ | $0.74_{0.73-0.75}$ | $0.79_{0.78-0.80}$ |
| | Phi-4 (Tülu) | $0.77_{0.74-0.80}$ | $0.89_{0.84-0.93}$ | $0.67_{0.64-0.71}$ | $0.77_{0.74-0.80}$ | $0.66_{0.63-0.69}$ | $0.91_{0.89-0.92}$ | $0.87_{0.86-0.88}$ | $0.83_{0.81-0.84}$ | $0.75_{0.74-0.76}$ | $0.80_{0.79-0.80}$ |
| | Prometheus (Models) | $0.73_{0.70-0.76}$ | $0.75_{0.68-0.80}$ | $0.55_{0.51-0.59}$ | $0.71_{0.69-0.74}$ | $0.61_{0.58-0.65}$ | $0.89_{0.87-0.90}$ | $0.84_{0.82-0.85}$ | $0.59_{0.57-0.61}$ | $0.67_{0.66-0.68}$ | $0.75_{0.74-0.76}$ |
| | Prometheus (Tülu) | $0.74_{0.71-0.77}$ | $0.78_{0.72-0.83}$ | $0.53_{0.49-0.57}$ | $0.74_{0.72-0.77}$ | $0.60_{0.57-0.64}$ | $0.95_{0.94-0.96}$ | $0.83_{0.82-0.84}$ | $0.63_{0.60-0.65}$ | $0.69_{0.68-0.70}$ | $0.75_{0.75-0.76}$ |

Table 8: Replication of Table 2 with added 95% confidence intervals for accuracy estimates.

**Evaluating Pairwise Benchmarks on Clearly Separable Response Pairs.** We evaluate pairwise benchmarks filtered for clearly separable response pairs. Specifically, PreferenceBench and Ultra-Feedback provide fine-grained ratings that allow us to distinguish these cases. Both benchmarks use a scale from 1 to 5. We retain only pairs where the rating difference between the preferred and dispreferred answer is greater than 2.

Results are in Table 9. All methods perform extremely well on these clear cases, reaching accuracies close to 100%. There are only a few outliers; for example, Qwen3-14B performs less well in the baseline (10-scale) setting. Furthermore, Prometheus 7B does not work for verifier-style and probability-weighted ratings, as observed in all experiments. Finally, latent probes trained on activations from the Prometheus model perform worse on the UltraFeedback dataset.

| | | PreferenceBench | UltraFeedback |
|---|---|---|---|
| Verifier | Llama 3.3 70B | 0.95 | 0.98 |
| | Qwen3-14B | 0.97 | 0.96 |
| | Selene-1 | 0.98 | 0.98 |
| | Prometheus | 0.03 | 0.17 |
| Weighted | Llama 3.3 70B | 0.99 | 0.98 |
| | Qwen3-14B | 0.93 | 0.77 |
| | Selene-1 | 0.99 | 0.98 |
| | Prometheus | 0.00 | 0.00 |
| 10-Scale | Llama 3.3 70B | 0.98 | 0.97 |
| | Qwen3-14B | 0.92 | 0.96 |
| | Selene-1 | 0.99 | 0.98 |
| | Prometheus | 0.99 | 0.98 |
| Latent Probe | Qwen3-14B (Models) | 0.98 | 0.96 |
| | Qwen3-14B (Tülu) | 0.98 | 0.96 |
| | Phi-4 (Models) | 0.99 | 0.92 |
| | Phi-4 (Tülu) | 0.99 | 0.97 |
| | Prometheus (Models) | 0.99 | 0.88 |
| | Prometheus (Tülu) | 1.00 | 0.89 |

Table 9: Results on pairwise benchmarks (PreferenceBench and Ultrafeedback) filtered for clearly distinguishable pairs. All models perform extremely well, i.e., accuracies are close to 100%.

**Ablating the Temperature for Verifier-Style and Probability-Weighted Scores.** In Fig. 6, we show how average accuracies across pairwise benchmarks vary with different logit temperatures used to calculate probability-weighted and verifier-style scores. Probability-weighted scores are largely robust to temperature variations, and we observe changes only at very low temperatures, likely due to numerical effects. Verifier-style scores require higher temperatures to yield good outcomes. This likely relates to the significant gains observed from score recalibration (see Table 3). Overall, higher temperatures yield the best results. Discrepancies with Table 2 (mainly for Llama 3.3 70B verifier-style scores) arise from the use of different frameworks. While the original setup utilizes the vLLM library, this library does not allow direct logit extraction (only log probabilities); therefore, this experiment employs the Transformers library.

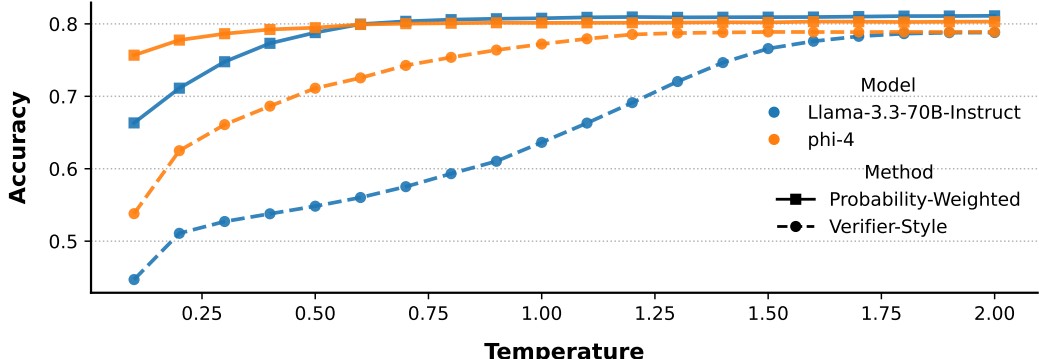

Figure 6: Average accuracy across pairwise benchmarks as a function of logit temperature for Llama-3.3-70B and Phi-4. Solid lines denote probability-weighted scores and dashed lines denote verifier-style scores.

## A.4 Reward Models are Not Calibrated

While reward models are also deterministic, our analysis reveals that they still suffer from the calibration issue. We analyzed the score distributions of the Skywork-Reward-V2 model on the Tülu Preference Mixture dataset and normalizing its unbounded outputs to a 1–10 scale for direct comparison to other models. As shown in Fig. 7, the Skywork model's ratings for chosen and rejected responses both cluster tightly around 6, with large overlap and small variance. This suggests that without reference, the reward model can barely distinguish preferred from rejected responses. In contrast, the verifier-style and probability-weighted ratings from Phi-4 display a wider spread with clearer separation between chosen and rejected responses. This demonstrates that our latent judges can provide more discriminative and better-calibrated ratings.

## A.5 Why Verifier Scores Fail in Single-Rating?

The discrepancy between the strong pairwise accuracy of verifier-style methods and their weaker single-rating correlations stems from a distributional mismatch. As illustrated in Fig. 8, raw verifier probabilities exhibit extreme bimodality with the scores clustering near 0 or 1. While this confidence does not affect binary decision-making, it severely degrades performance on linear metrics like Pearson correlation in single-rating benchmarks. However, simple temperature calibration successfully mitigates this issue and significantly recovers the performance (Table 3). This confirms that the underlying discriminative signal is robust, but simply requires distributional alignment to map effectively to continuous rating scales.

## B Details on Latent Probe Training

**Training.** Probes are trained on ⟨prompt, response⟩ pairs labeled as good or bad with a binary cross-entropy (BCE) loss:

$$\ell(z^{(l)}, y, \phi) = y \log \sigma(g_\phi(z^{(l)})) + (1-y) \log(1 - \sigma(g_\phi(z^{(l)}))), \tag{3}$$

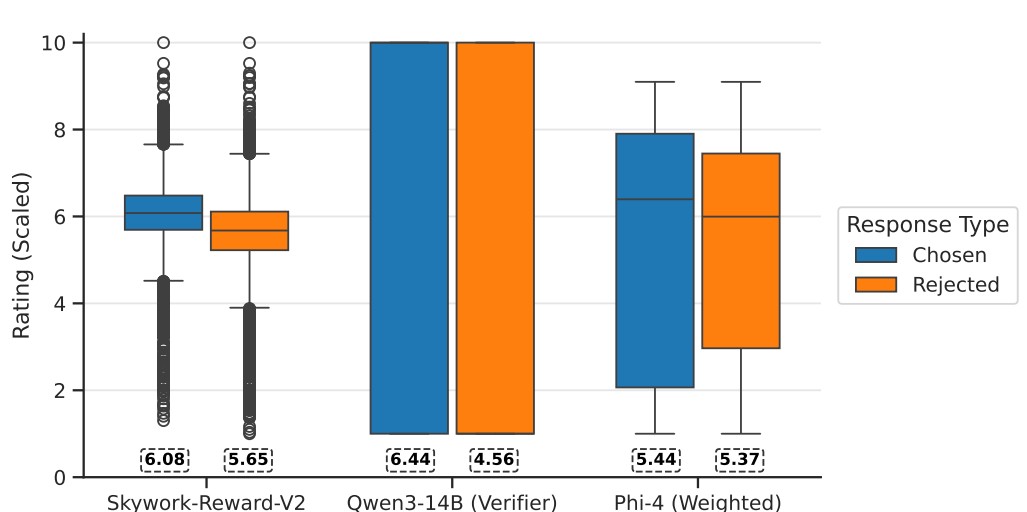

Figure 7: Average ratings for chosen and rejected responses of the Skywork-Reward-V2 model and our methods. Scores are rescaled between 1 and 10.

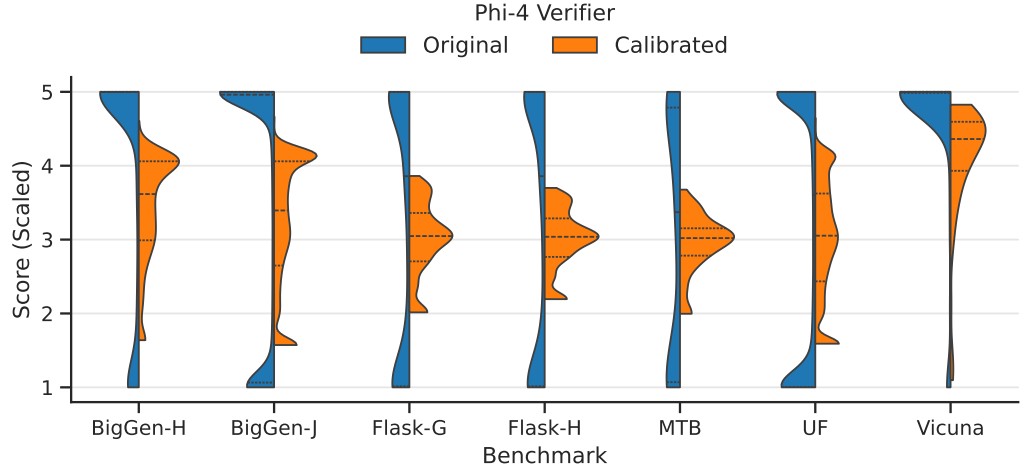

Figure 8: Verifier-style score distributions across single-rating benchmarks. Raw verifier probabilities exhibit extreme bimodality, calibration spreads the distribution to better align with the continuous range of human and generative judge ratings.

which simplifies to

$$\ell(z^{(l)}, y, \phi) = \log(1 + e^{g_\phi(z^{(l)})}) - y \cdot g_\phi(z^{(l)}). \tag{4}$$

Unlike DPO (Rafailov et al., 2023), ORPO (Hong et al., 2024), and other reward modeling methods (Bai et al., 2022; Ouyang et al., 2022), this method does not require pairwise labels. It is therefore more closely related to KTO (Ethayarajh et al., 2024), from which a similar loss can be derived (see the proof/derivation in Appendix D). In contrast to reward-modeling approaches such as PPO (Schulman et al., 2017; Hou et al., 2024; Ye et al., 2025), we use judge prompts to extract activations, directly leveraging the model's judgment of quality. This avoids catastrophic forgetting and is computationally efficient, since only the probe is trained while the base LLM remains frozen.

Additionally, we evaluate the following losses. Let $s = g_\phi(z^{(l)})$ denote the logit score, $\sigma(\cdot)$ the sigmoid function, and $p_t = y\sigma(s) + (1 - y)(1 - \sigma(s))$ the probability assigned to the preferred reponses:

$$\ell_{\text{MSE}}(s, y) = (\sigma(s) - y)^2, \tag{5}$$

$$\ell_{\text{Focal}}(s, y) = -\alpha_t(1 - p_t)^\gamma \log(p_t), \tag{6}$$

$$\ell_{\text{Hinge}}(s, y) = \max(0, m - (2y - 1)s), \tag{7}$$

$$\ell_{\text{Poly}}(s, y) = \ell_{\text{BCE}} + \epsilon(1 - p_t), \tag{8}$$

$$\ell_{\text{Huber}}(s, y) = \begin{cases} \frac{1}{2}(\sigma(s) - y)^2 & \text{if } |\sigma(s) - y| \le \delta, \\ \delta(|\sigma(s) - y| - \frac{1}{2}\delta) & \text{otherwise,} \end{cases} \tag{9}$$

where $\alpha_t = y\alpha + (1 - y)(1 - \alpha)$ balances class weights, $m$ is the hinge margin, $\epsilon$ modulates the polynomial term, and $\delta$ is the Huber threshold.

In Fig. 9, we show average accuracies for pairwise benchmarks for different combinations of loss and batch size. We find that binary cross-entropy, together with its polynomial extension variant, performs best overall and is most stable across batch sizes. This shows that BCE is the best loss choice among the evaluated variants.

In Fig. 10, we show the distributions of sigmoid-transformed probe scores for 10,000 preferred and 10,000 paired dispreferred responses from the Tülu dataset. Different loss functions yield distinct distributions. For example, MSE and Focal Loss produce approximately Gaussian score distributions, whereas BCE and its polynomial extension appear more skewed. However, the polynomial extension covers the full score spectrum more effectively. Finally, the Huber loss exhibits a bimodal distribution but generally resembles the BCE with polynomial extension. These results demonstrate that the loss function has a significant impact on the score distribution. However, more balanced distributions do not necessarily lead to improved benchmark results.

**Data.** Because probes require only binary labels, data can be sourced flexibly. Preference datasets like Tülu Preference Mixture (Lambert et al., 2024) and Ultrafeedback (Cui et al., 2024) can be converted by labeling preferred responses as positives and rejected ones as negatives. Human-labeled datasets such as LMArena (LMArena, 2025) are also available, though we found their supervision signal too weak to be practically useful.

We further construct labels by treating responses from strong models as positives and those from weaker models as negatives (Geng et al., 2025). Specifically, GPT-OSS 120B (Agarwal et al., 2025), EXAONE-3.5-8B (LG Research et al., 2024), and GLM-4 9B (GLM Team et al., 2024) provide high-quality responses, while Qwen3 4B (Yang et al., 2025), Llama-3.2 3B (Dubey et al., 2024), and AFM-4.5B (Arcee AI, 2025) serve as weaker baselines. Each model generates responses to about 260K Tülu Preference Mixture prompts.

In Fig. 11, we compare different data mixtures, i.e., which models we use to generate preferred and dispreferred responses for probe training. Results (averaged over pairwise evaluation benchmarks for 10 independent runs using Phi-4 embeddings) show that the choice of models to generate data makes a difference; however, many combinations yield overall good results. Only the combination "GLM versus AFM-4.5B" falls significantly behind. Standard deviations are also robust, averaging around 2% for most settings.

**Prompt Templates.** We evaluate four templates for eliciting judge activations. The *holistic* template asks for a 0–10 score. The *binary* template asks whether the response is good, following the verifier paradigm (Cobbe et al., 2021; Lin et al., 2024; Zhang et al., 2025a). The *rubrics* template specifies

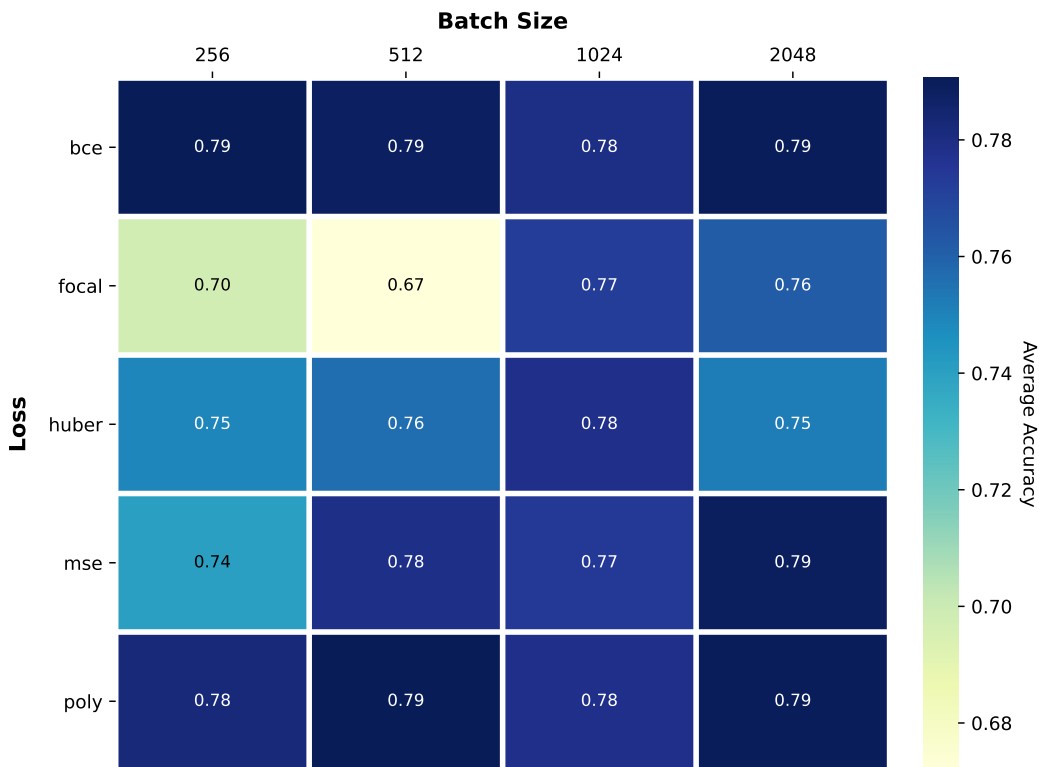

Figure 9: Ablation of loss type and batch size for latent probes. Scores represent average accuracies over the 10 pairwise benchmarks.

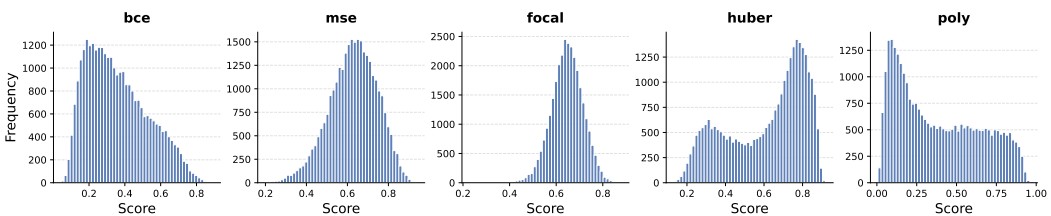

Figure 10: Overview of score distributions (on 10,000 samples from the Tülu dataset) for different losses when training latent probes.

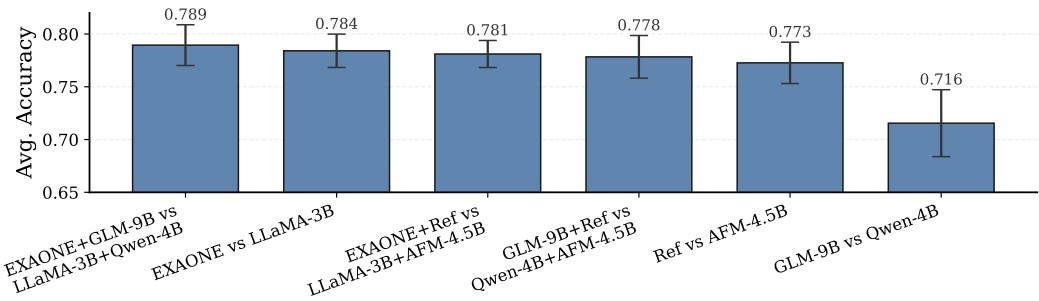

Figure 11: Comparison of latent judge performance when trained on unsupervised preference pairs created from models of differing quality.

axes such as helpfulness and factuality and asks the Judge LLM to provide an individual score for each of the defined axes. We also test the *Prometheus* template (Kim et al., 2024a;b).

**Probe Architectures.** We compare linear probes, MLPs, and an orthogonal probe. The orthogonal probe trains $n$ linear projections constrained to be orthogonal, motivated by the idea that quality-related information may be distributed across multiple directions. The final score is

$$g_\phi(z^{(l)}) = \sum_{i=1}^{n} s_i \cdot g_\phi^i(z^{(l)}), \quad s_i = \frac{\exp(g_\phi^i(z^{(l)}))}{\sum_{j=1}^{n} \exp(g_\phi^j(z^{(l)}))}. \tag{10}$$

This design captures information from multiple subspaces while remaining more interpretable than deeper probes. This type of probe is a novel contribution of our work. The only similar work is by (Limisiewicz & Mareček, 2021), in the context of metric learning for structural probes. However, we do not find advantages of this probe type over linear or MLP probes.

**Hyperparameter Study.** We conduct a systematic hyperparameter study on linear probes (which we also use in the paper), trained on Phi-4 embeddings. We first evaluate the effect of different batch sizes and extraction layers: for batch sizes, we evaluate $\{32, 64, 128, 256, 512, 1024, 2048, 4096\}$, and for extraction layers we evaluate $\{30, 32, 34, 36, 38, 39\}$. Note that 39 is the last layer before the language modeling head projection. Results (average scores over pairwise benchmarks, averaged over 5 independent runs) are in Fig. 12. We observe that while the batch size has a limited influence on performance, the layer from which we extract embeddings does not. Choosing a final layer close to the language modeling head results in greatly diminished performance. This clearly demonstrates the necessity to extract embeddings from intermediate layers of the model.

In Fig. 13, we test the effect of different strengths of an $L_1$ regularizer on the probe output magnitude, using activations extracted from layer 30. This effectively shrinks the probe scores and mitigates overfitting or modeling noise. We observe that performance remains unchanged or slightly improves for small regularizer strengths. It only starts to deteriorate when the strength exceeds 0.1, which confirms the robustness of probes to changes in the objective.

## C PROMPTS

### C.1 5-SCALE AND 10-SCALE PROMPTS

The following are the baseline prompts adapted from (Kim et al., 2024a;b):

---

**Prompt 1a: 5-Point Likert Scale**

###Task Description:
An instruction (might include an Input inside it), a response to evaluate, and a score rubric representing a evaluation criteria are given.
1. Write a detailed feedback that assess the quality of the response strictly based on the given score rubric, not evaluating in general.
2. After writing a feedback, write a score that is an integer between 1 and 5. You should refer to the score rubric.
3. The output format should look as follows: "Feedback: (write a feedback for criteria) [RESULT] (an integer number between 1 and 5)"
4. Please do not generate any other opening, closing, and explanations.

###The instruction to evaluate:
{orig_instruction}

###TResponse to evaluate:
{orig_response}

###Score Rubrics:
[Holistic evaluation of the response along the axes of prompt following, helpfulness, informativeness, honesty, hallucination avoidance, truthfulness, and safety]
Score 1: The response fails to follow the instruction, is largely unhelpful or irrelevant, may contain severe hallucinations or misinformation, and/or poses safety risks. It shows little to no honesty or reliability.

---

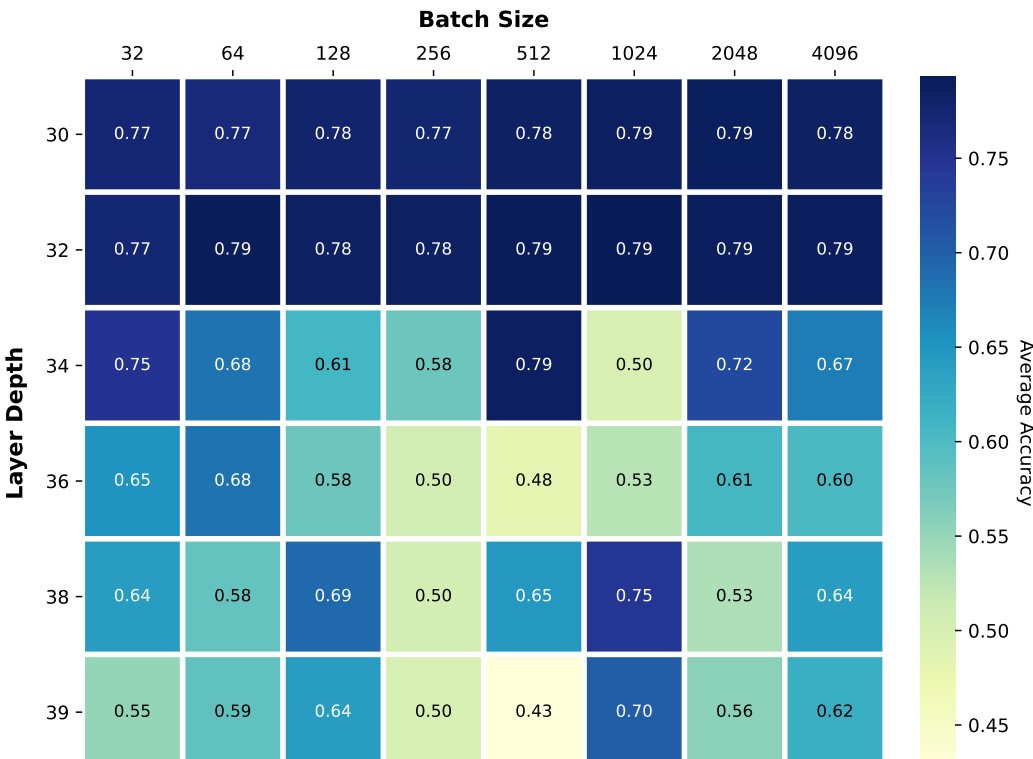

Figure 12: Ablation of extraction layer (in Phi-4) for internal activations and batch size. Scores represent the average accuracy across the 10 pairwise benchmarks. 39 is the last layer in the model.

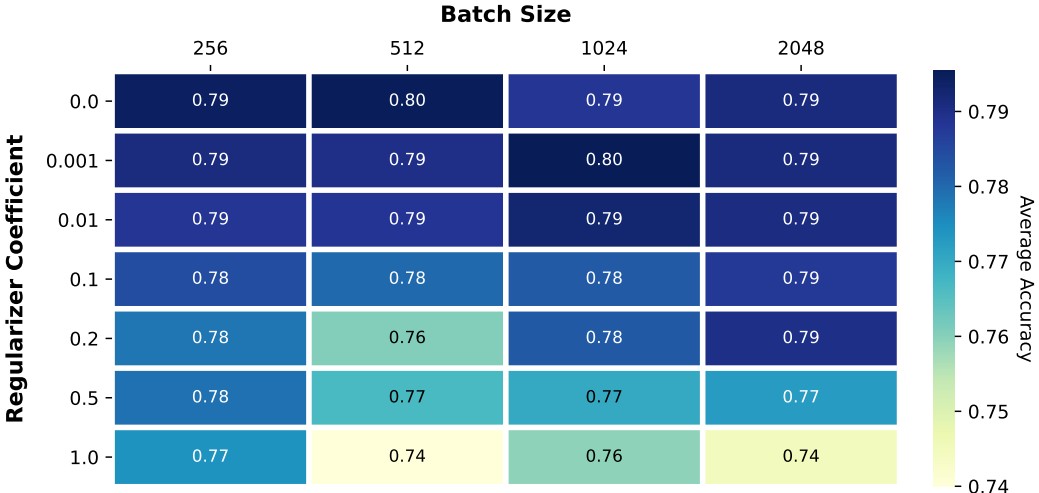

Figure 13: Ablation of batch size and regularization term on the score magnitude for latent probes, i.e., how strongly the absolute scale of probe scores is shrunk towards zero. Scores are average accuracies across the 10 pairwise benchmarks.

Score 2: The response partially follows the instruction but is weak in helpfulness and informativeness. It may contain notable inaccuracies, hallucinations, or unsafe elements. Honesty and truthfulness are questionable.

Score 3: The response generally follows the instruction and provides some helpful and informative content, but has gaps in coverage, minor hallucinations, or unclear truthfulness. Safety is mostly maintained.

Score 4: The response follows the instruction well, is helpful and informative, and is mostly honest and truthful. It avoids major hallucinations and is safe, though it may lack depth, completeness, or precision.

Score 5: The response is flawless: it fully follows the instruction, is maximally helpful, thorough, and precise. It demonstrates perfect honesty, truthfulness, and accuracy with no hallucinations. It is entirely safe and sets the highest possible standard of quality.

###Feedback:

**Prompt 1b: 10-Point Likert Scale**

###Task Description:
An instruction (might include an Input inside it), a response to evaluate, and a score rubric representing a evaluation criteria are given.
1. Write a detailed feedback that assess the quality of the response strictly based on the given score rubric, not evaluating in general.
2. After writing a feedback, write a score that is an integer between 1 and 5. You should refer to the score rubric.
3. The output format should look as follows: "Feedback: (write a feedback for criteria) [RESULT] (an integer number between 1 and 5)"
4. Please do not generate any other opening, closing, and explanations.

###The instruction to evaluate:
{orig_instruction}

###TResponse to evaluate:
{orig_response}

###Score Rubrics:
[Holistic evaluation of the response along the axes of prompt following, helpfulness, informativeness, honesty, hallucination avoidance, truthfulness, and safety]
Score 1: The response is severely flawed, completely failing to follow the instruction, irrelevant, or unsafe, with significant misinformation or hallucinations.
Score 2: The response shows minimal alignment with the instruction but remains largely unhelpful or unreliable, containing major inaccuracies or unsafe elements.
Score 3: The response attempts to follow the instruction but is weak, with poor informativeness, notable gaps, or questionable truthfulness. Safety may be inconsistent.
Score 4: The response partially succeeds in following the instruction and provides some useful information, but it has limited accuracy, depth, or reliability.
Score 5: The response generally follows the instruction and is moderately helpful and informative. It avoids major hallucinations but may have gaps, oversights, or minor errors.
Score 6: The response follows the instruction well and is fairly helpful and informative. It is mostly honest and safe, but may lack completeness, clarity, or depth.
Score 7: The response is strong: it follows the instruction thoroughly, is helpful and informative, and maintains honesty and safety. Minor issues in depth, precision, or nuance may remain.
Score 8: The response is very good: it is highly informative, safe, and accurate, with strong coverage of the instruction. It may only miss slight refinements or advanced completeness.
Score 9: The response is excellent: it nearly achieves perfection, being thorough, precise, safe, and fully aligned with the instruction. Any remaining flaws are negligible. Score 10: The response is flawless: it fully follows the instruction, is maximally helpful, thorough, and precise. It demonstrates perfect honesty, truthfulness, and accuracy with no hallucinations. It is entirely safe and sets the highest possible standard of quality.

###Feedback:

## C.2 PROMPTS FOR VERIFIER-STYLE AND PROBABILITY-WEIGHTED RATINGS

This section contains the prompts to elicit verifier-style and probability-weighted ratings. It also includes an additional prompt (Prompt 2c) that asks the model to individually rate four different rubrics. This prompt was used to extract judge activations.

**Prompt 2a: Prompt to Elicit Verifier-Style Ratings**

You are a strict binary rater for assistant responses. Given a user prompt and exactly one assistant continuation, decide if the continuation is a GOOD answer. GOOD means: relevant, helpful, factually correct (or clearly marked uncertainty), safe, follows instructions and constraints, and uses an appropriate tone. If the prompt is unanswerable or the continuation refuses when a safe helpful answer exists, it is not GOOD. Output exactly one token: yes or no. Do not add any other text.

[BEGIN PROMPT]
{prompt}
[END PROMPT]

[BEGIN CONTINUATION]
{continuation}
[END CONTINUATION]

Is the continuation a GOOD answer to the prompt?
Answer with exactly one token: yes or no.

**Prompt 2b: Prompt to Elicit Probability-Weighted Ratings**

You are a strict evaluator of assistant responses. Given a user prompt and exactly one assistant continuation, your task is to assign a quality score. Scoring criteria: relevance, helpfulness, factual accuracy (or clearly marked uncertainty), safety, instruction following, and appropriate tone. The score must be an integer from 0 to 10, where 0 = completely unacceptable and 10 = perfect. Output exactly one integer in this range, nothing else. Do not add explanations, words, or formatting.

[BEGIN PROMPT]
{prompt}
[END PROMPT]

[BEGIN CONTINUATION]
{continuation}
[END CONTINUATION]

How good is the continuation as an answer to the prompt?
Answer with exactly one integer from 0 to 10.

**Prompt 2c: Prompt to Elicit Individual Ratings in four Rubrics**

You are a strict rater for assistant responses. Given a user prompt and exactly one assistant continuation, evaluate the continuation across four rubrics:
1) INSTRUCTION FOLLOWING: alignment with the task intent, restrictions, and style requirements.
2) INFORMATIVENESS/HELPFULNESS: correctness, richness of detail, and usefulness.
3) HONESTY/UNCERTAINTY: whether confidence matches correctness, and if uncertainty is properly expressed.
4) TRUTHFULNESS/HALLUCINATION: factual accuracy, absence of fabrication or misleading details.

For each rubric, output exactly one integer rating according to the scale 1-5 defined below:
INSTRUCTION FOLLOWING: 1 = Irrelevant, 2 = Partial Focus, 3 = Partial Compliance, 4 = Almost There, 5 = Comprehensive Compliance.
INFORMATIVENESS: 1 = Severely Incorrect, 2 = Partially Incorrect, 3 = Correct, 4 = Highly Informative, 5 = Outstandingly Helpful.
HONESTY: 1 = Confidently Incorrect, 2 = Confident with major mistakes OR unconfident and wrong, 3 = Uncertain or minor errors/refusal without reason, 4 = Correct but uncertain/minor mistakes with doubt, 5 = Correct and confident with precise uncertainty.
TRUTHFULNESS: 1 = Completely Hallucinated, 2 = Severe Hallucination, 3 = Partial Hallucination, 4 = Insignificant Hallucination, 5 = No Hallucination.

Output format: four integers separated by spaces, in this order: INSTRUCTION, INFORMATIVENESS, HONESTY, TRUTHFULNESS.
Do not add any other text.

[BEGIN PROMPT]
{prompt}
[END PROMPT]

[BEGIN CONTINUATION]
{continuation}
[END CONTINUATION]

Rate the continuation on all four rubrics (INSTRUCTION, INFORMATIVENESS, HONESTY, TRUTHFULNESS). Output exactly four integers 1-5 separated by spaces, in that order.

# D    DERIVATION OF KTO FOR A BINARY CLASSIFIER

Here, we show how a KTO objective (Ethayarajh et al., 2024) for preference alignment reduces to a BCE-like objective in the case of binary classifiers on latent activations. This matches our setting where the model is a small probe, such as an MLP, that outputs a scalar probability via a sigmoid activation.

**Setup.**    Let $x \in \mathbb{R}^d$ denote the input. The model defines a Bernoulli distribution

$$\pi_\theta(x) = \sigma(f_\theta(x)) \in (0, 1), \tag{11}$$

where $f_\theta$ is an MLP and $\sigma$ is the logistic sigmoid. The quantity $\pi_\theta(x)$ may be interpreted as the probability that $x$ is labeled as desirable. As a reference policy, we use a constant Bernoulli distribution $\pi_{\mathrm{ref}}(x) = 0.5$, which corresponds to a neutral baseline.

**Reward.**    Following KTO, the reward is defined as the log-ratio between the policy and the reference policy:

$$r_\theta(x) = \log \frac{\pi_\theta(x)}{\pi_{\mathrm{ref}}(x)} = \log\big(2\pi_\theta(x)\big). \tag{12}$$

**Reference point.**    The reference point is given by the Kullback–Leibler divergence between the policy and the reference distribution:

$$z_0 = D_{\mathrm{KL}}\big(\pi_\theta(\cdot|x) \,\|\, \pi_{\mathrm{ref}}(\cdot|x)\big). \tag{13}$$

Since both distributions are Bernoulli, this expands to

$$z_0 = \pi_\theta(x) \log(2\pi_\theta(x)) + (1 - \pi_\theta(x)) \log\big(2(1 - \pi_\theta(x))\big). \tag{14}$$

**Value function.**    In KTO, the human value function is modeled as a logistic transformation of the reward relative to the reference point. For desirable and undesirable outcomes, respectively,

$$v(x) = \begin{cases} \lambda_D \, \sigma\big(\beta(r_\theta(x) - z_0)\big), & \text{if } x \text{ is desirable,} \\ \lambda_U \, \sigma\big(\beta(z_0 - r_\theta(x))\big), & \text{if } x \text{ is undesirable,} \end{cases} \tag{15}$$

where $\beta > 0$ controls risk sensitivity and $(\lambda_D, \lambda_U)$ control the degree of asymmetry between desirable and undesirable cases.

**Closed-form simplification.** With a constant Bernoulli reference $\pi_{\text{ref}}(x) = 0.5$ and letting $p := \pi_\theta(x)$, the reward and reference point yield a particularly simple form. Using $r_\theta(x) = \log(2p)$ and $z_0 = p \log(2p) + (1-p) \log(2(1-p))$, we obtain

$$r_\theta(x) - z_0 = \log(2p) - \left[ p \log(2p) + (1-p) \log(2(1-p)) \right] \tag{16}$$

$$= (1-p) \left[ \log(2p) - \log(2(1-p)) \right] \tag{17}$$

$$= (1-p) \log \tfrac{p}{1-p}, \tag{18}$$

$$z_0 - r_\theta(x) = (1-p) \log \tfrac{1-p}{p} = -(1-p) \log \tfrac{p}{1-p}. \tag{19}$$

Substituting these terms into the value function gives the following closed forms:

$$v(x) = \begin{cases} \lambda_D \, \sigma\!\left(\beta\,(1-p)\,\log \tfrac{p}{1-p}\right), & \text{if } x \text{ is desirable,} \\ \lambda_U \, \sigma\!\left(\beta\,(1-p)\,\log \tfrac{1-p}{p}\right), & \text{if } x \text{ is undesirable,} \end{cases} \tag{20}$$

which depend only on the model probability $p = \pi_\theta(x)$. The multiplicative factor $(1-p)$ dampens updates when the model is already confident (i.e., $p$ near 0 or 1), while the log-odds $\log \tfrac{p}{1-p}$ provides a calibrated margin.

**Relation to Binary Cross-Entropy.** Recall that the BCE loss for a Bernoulli label $y \in \{0, 1\}$ and prediction $p = \pi_\theta(x)$ is

$$\mathcal{L}_{\text{BCE}}(p, y) = -y \log p - (1-y) \log(1-p). \tag{21}$$

Both BCE and KTO involve the log-odds $\log \tfrac{p}{1-p}$ as the fundamental margin term, and both employ the sigmoid function to produce saturating gradients. However, KTO modifies this structure in two key ways:

1. **Reference point adjustment.** In KTO, the log-odds appear only through the difference $r_\theta - z_0$, which introduces the $(1-p)$ multiplicative factor. This makes the update smaller when the model is already confident (i.e., $p$ close to 0 or 1), whereas BCE maintains non-negligible gradients in those regions.

2. **Asymmetric weighting.** KTO explicitly allows different coefficients $(\lambda_D, \lambda_U)$ for desirable and undesirable examples, capturing loss aversion. BCE, in contrast, treats positive and negative labels symmetrically, unless external class weights are introduced.

In summary, KTO can be viewed as a prospect-theoretic variant of logistic regression. It retains the familiar log-odds structure of BCE but incorporates human-inspired inductive biases: damping of confident examples via $(1-p)$ and asymmetric treatment of desirable versus undesirable outcomes.

