# OpenReview forum: "Reference-Free Rating of LLM Responses via Latent Information"
_ICLR.cc/2026/Conference — Submitted to ICLR 2026_

### Official Review · Reviewer_XxtF · 2025-10-24

**Soundness:** 3
**Presentation:** 4
**Contribution:** 3
**Rating:** 6
**Confidence:** 4

**Summary:**

This paper challenges the reliability of the common practice of using LLMs to assign single-response, reference-free Likert-scale scores to text. The authors identify two primary weaknesses with this approach: **instability** (scores vary with stochastic decoding) and **poor calibration** (scores are compressed near the top of the scale, leading to frequent ties and low discriminability). To address these issues, this paper proposes and evaluates **Latent Judges**, a set of three methods that derive deterministic, fine-grained scores from internal model signals rather than generated tokens.

The authors present an evaluation spanning a wide range of both pairwise and single-rating benchmarks. Their results show that latent approaches, especially probability-weighted and verifier-style methods, consistently match or outperform standard prompting baselines on pairwise accuracy. While results on single-rating correlation tasks are more mixed, probability-weighted scores remain strong. Overall, the paper shows that leveraging latent information yields more stable and discriminative signals for reference-free evaluation, which could benefit applications such as Best-of-N selection and model routing.

**Strengths:**

* Originality: While probing and using logits have been explored before, as I know, this work is the first to systematically identify the instability and calibration issues of single-response LLM judging and then propose and validate a suite of "Latent Judge" methods as a direct solution. This reframing of the problem is a novel conceptual contribution.
* Quality: The paper demonstrates good quality through its methodologically rigorous and extensive empirical evaluation. The problem is diagnosed systematically and quantified with clear metrics across a large dataset. The evaluation of the proposed solutions is conducted across 10 pairwise and 5 single-rating benchmarks, providing robust and comprehensive evidence for the claims.
* Clarity: The paper's clarity is a significant strength. The core problem is visualized in Figures 1 and 2, which clearly show the inconsistency and score compression. The proposed methods are explained concisely. The extensive results are well presented in organized tables for direct comparison.
* Significance: The work is of considerable significance to the MLOps and evaluation communities. It addresses a practical and increasingly common failure mode in LLM evaluation. By providing a set of solutions—some of which are training-free and easy to implement (like probability-weighted scores)—the paper offers an immediate and actionable path for practitioners to improve the reliability and discriminative power of their evaluation pipelines for tasks like Best-of-N selection, routing, and multi-teacher distillation.

**Weaknesses:**

* Limited novelty: The core ideas of using next-token probabilities or training probes on internal model states have been explored in other contexts (especially for explainability). The main novelty lies in the application and systematic study, so work feels more incremental than paradigm-shifting.
* Degraded performance on single-rating benchmarks: While the latent methods excel at pairwise comparisons, their advantage is less clear on single-rating correlation tasks. The paper acknowledges this but could benefit from a deeper analysis of why this discrepancy exists.
* Effectiveness of probes is data-dependent: The success of the latent probe method is shown to be dependent on the training data source (Tülu-trained probes outperform model-generated-data probes). This highlights a dependency that complicates its application as a truly general, "reference-free", "off-the-shelf" solution.

**Questions:**

* Your results show that fine-tuned judges like Prometheus can fail entirely on alternative prompting formats (verifier and probability-weighted). This suggests a form of "prompt overfitting." Do you believe this is a fundamental limitation of fine-tuning for judge models, or could it be addressed with a more diverse fine-tuning data mixture that includes different evaluation formats?
* The performance of verifier-style ratings is strong in pairwise tasks but weaker in single-rating correlation tasks. Your analysis suggests this is due to scores concentrating near 0 or 1. Could this be mitigated by temperature scaling or other calibration methods applied to the yes/no logits before calculating the probability?
* For the latent probes, you train on the residual stream activations. Did you experiment with probing activations from different layers? Is there a reason to believe the final layer is the most informative for this task, or could earlier layers capture different or more robust signals of quality?

---

> ### Author Response · Authors · 2025-12-02
> **Author Response to Reviewer XxtF (1/2)**
>
> We thank the reviewer for the positive feedback, highlighting our “novel conceptual contribution”, “rigorous and extensive empirical evaluation”, and the “clarity” and “significance” of our work. We are happy the reviewer sees our work as “of considerable significance to the MLOps and evaluation communities”. Below, we respond to the individual concerns mentioned by the reviewer.
>
> ---
>
> > **(W1)** Limited novelty: The core ideas of using next-token probabilities or training probes on internal model states have been explored in other contexts (especially for explainability). The main novelty lies in the application and systematic study, so work feels more incremental than paradigm-shifting.
>
> We acknowledge that prediction probabilities are used in other contexts. As the reviewer points out, our contribution evaluates different methods based on latent information for LLM-as-a-judge and demonstrates consistent improvements over the baseline across a wide range of settings. We maintain that using latent information rather than model generations represents a paradigm shift. This approach enables new applications that improve response rating. In our revision, we included a new statement in the related work section (Sec. 2, “Latent Probing”) to position our contributions with respect to other applications of latent probing.
>
> ---
>
> > **(W2)** Degraded performance on single-rating benchmarks: While the latent methods excel at pairwise comparisons, their advantage is less clear on single-rating correlation tasks. The paper acknowledges this but could benefit from a deeper analysis of why this discrepancy exists.
>
> Thank you for pointing this out. In our revision, we included a more detailed analysis of why verifiers fail in single-rating benchmarks in Appendix A.4. As illustrated in Fig. 8, raw verifier probabilities exhibit extreme bimodality with the scores clustering near 0 or 1. While this confidence does not affect binary decision-making, it severely degrades performance on linear metrics like Pearson correlation in single-rating benchmarks. However, by doing simple temperature calibration (as the Reviewer also suggested), we successfully mitigate this issue and significantly recover the performance (Table 3). This confirms that the underlying discriminative signal of verifiers is in fact robust, but simply requires distributional alignment to map effectively to continuous rating scales.
>
> ---
>
> > **(W3)** Effectiveness of probes is data-dependent: The success of the latent probe method is shown to be dependent on the training data source (Tülu-trained probes outperform model-generated-data probes). This highlights a dependency that complicates its application as a truly general, "reference-free", "off-the-shelf" solution.
>
> This is a good observation. In our paper, we use two setups: (1) Training the probe on publicly available GPT-4 rated data, i.e., the Tülu Preference Mixture; (2) Training on responses from pairs consisting of a stronger and a weaker model. Here, our aim is to evaluate how big the gap is between “ground-truth” GPT-rated data and unsupervised data from model pairs. Option (1) is meant as an upper bound, however we would still like to stress that we only used openly available, i.e. already existing, data without generating new preference pairs from GPT judgements. While it is true, as the reviewer notices, that Tülu-trained probes outperform probes trained on unsupervised data, we want to stress that the performance gap between option (1) and option (2) is actually very small in Table 4 and 5, meaning that we can replace GPT-rated pairs with unsupervised pairs with little or no loss in performance.
>
> For option (2), we also observe that indeed the selected model pairs have influence on the performance. We evaluated several combinations of models and included this evaluation in the revision in Appendix B (“Data”, Fig. 11). There, we show that several data mixes yield comparable performance and only one evaluated setting deviates. This shows that care must be taken when selecting models, but already reasonable comparisons, for example EXAONE-3.5-7.8B vs. Llama3 3B closely follows the performance of GPT-rated pairs. From this evaluation, we conclude that it is not very hard to find model pairs to successfully generate unsupervised pairs of preferred and dispreferred responses. The only condition is that the performance gap between the models is sufficiently large. However, such constraints are expected for unsupervised methods and they do not seriously limit the impact or applicability of our contributions.

---

> > ### Author Response · Authors · 2025-12-02
> > **Author Response to Reviewer XxtF (2/2)**
> >
> > ---
> >
> > > **(Q1)** Your results show that fine-tuned judges like Prometheus can fail entirely on alternative prompting formats (verifier and probability-weighted). This suggests a form of "prompt overfitting." Do you believe this is a fundamental limitation of fine-tuning for judge models, or could it be addressed with a more diverse fine-tuning data mixture that includes different evaluation formats?
> >
> > Thank you for asking these questions. We observe this in the context of Prometheus, which in the context of verifier-style ratings or probability-weighted ratings often does not follow the prompt and is hence unusable for these settings. This is most likely an artifact of “prompt overfitting”, as the Reviewer observes. This underscores the importance of developing latent probing solutions, because we show that the performance can be recovered by using internal activations of the model, even when the output logits are not informative. At the same time, we agree that this problem is not universal to all models tuned specifically for LLM-as-a-judge applications. For example, Selene-1 doesn’t have this issue, which suggests their training recipe avoids this problem.
> >
> > ---
> >
> > > **(Q2)** The performance of verifier-style ratings is strong in pairwise tasks but weaker in single-rating correlation tasks. Your analysis suggests this is due to scores concentrating near 0 or 1. Could this be mitigated by temperature scaling or other calibration methods applied to the yes/no logits before calculating the probability?
> >
> > Thank you for this insightful suggestion. We have included the calibrated verifier-style scoring results in Table 3 and the discussion in Sec. 4.4 in our revision. We calibrated the verifier by extracting the logits of \enquote{yes} ($z_\text{yes}$) and applying a temperature-scaled sigmoid transformation $S_b = \sigma\left(\frac{z}{T}\right)$. For each model-dataset pair, we estimate the optimal temperature $T$ using a stratified 5\% data split and evaluate on the remaining.  $T$ is optimized via the L-BFGS-B algorithm (constrained to $T \in [1, 30]$) to minimize the Mean Squared Error (MSE) between the predicted ratings and the ground truth scores. As shown in Table 3, calibration consistently improved Pearson correlations across all models; for example, Qwen3-14B improved from 0.46 to 0.54 on average. Crucially, the calibrated verifier scores now match or nearly match the performance of the 10-scale baseline (e.g., Phi-4 achieves 0.59 in both settings), offering another strong alternative method.
> >
> > ---
> >
> > > **(Q3)** For the latent probes, you train on the residual stream activations. Did you experiment with probing activations from different layers? Is there a reason to believe the final layer is the most informative for this task, or could earlier layers capture different or more robust signals of quality?
> >
> > Yes, we experimented with activations from different layers. In fact, we already used intermediate layers, i.e. layer 31 (of 40) for Qwen3-14B and Phi-4, and layer 21 of 32 for Prometheus-7B. These yield superior performance compared to later layers, which we attribute to the richer information processed by LLMs in intermediate layers, while final layers are more specialized towards turning the abstract information into concrete language presentation. In our revision, we have included an ablation of the extraction layer for Phi-4 in Appendix B (“Hyperparameter Study”, Fig. 12).
> >
> > ---
> >
> > We would like to thank the reviewer once again for their positive evaluation of our work and insightful review. We have updated our paper to address all the concerns and questions mentioned.

---

### Official Review · Reviewer_bhBs · 2025-10-24

**Soundness:** 3
**Presentation:** 3
**Contribution:** 2
**Rating:** 6
**Confidence:** 3

**Summary:**

This paper analyzes key reliability gaps in existing LLM-as-a-Judge frameworks (how to produce consistent and calibrated responses for free-text responses) and proposes solutions guided by internal model signals (e.g., Probability-weighted ratings, Verifier-style probabilities, and Latent probes). Across a suite of pairwise and single-rating benchmarks, their proposed methods that rely on internal model signals outperform or match standard prompting in aligning with human-annotators (or stronger LLM judges). Results suggest that latent internal signals may be more stable and calibration-friendly for preference-based evaluation.

**Strengths:**

1.	Well-motivated and methodologically sound. The paper addresses a highly relevant problem—instability and calibration collapse in LLM-based evaluation—and proposes solutions (probability-weighted ratings and latent probes) that are both rigorous and intuitively grounded.
2.	Comprehensive benchmarking. The study systematically evaluates across ten preference datasets, covering diverse judge models and scoring regimes.
3.	Insightful empirical findings. The result that fine-tuned judgment models do not necessarily outperform general-purpose LLMs is particularly interesting and challenges common assumptions about specialized post-training.

**Weaknesses:**

1.	Novelty. While the latent probing approach is interesting, the “verifier-style” probability concept has been explored previously in domains like math/code. The paper should better articulate what is new or generalizable about extending this idea to preference judgment.
2.	Distinguishing LLM-as-a-Judge vs. Reward Models. The motivation would benefit from a clearer conceptual separation between these paradigms. For example, Best-of-N selection pipelines may use trained reward models rather than LLM judges. Including a SOTA reward model (e.g., top of RewardBench) (or justification for the choice of already-included models that are fine-tuned for judging) in the determinism and calibration analysis could reveal whether reward-model scores are inherently more deterministic.
3.	Comparative baseline. Table 1 currently benchmarks against GPT-4 judgments but not against post-trained reward models. Comparing agreement with a strong, fine-tuned reward model would add to the evaluation.
4.    Sensitivity to Temperature. The determinism experiment should test multiple temperature settings to confirm that observed variance is not an artifact of decoding randomness.
5.	Explanation of why instability occurs. The paper attributes inconsistency and poor calibration to discrete categories and stochastic decoding but does not elaborate. A dedicated section explaining the intuition behind these mechanisms would round out the storyline.
6.	Potential label noise. It would be interesting to replicate Table 4 using only clearly separable examples (e.g., where human preference strongly agrees across annotators) to test robustness (Frequent ties might indicate noise or low-signal regions in human-labeled preference data.)
7.	Please add confidence intervals for the mean estimates in Table 4.
8.	Generalizability of Latent Probes. The finding that probes trained on Tülu responses perform best raises concerns about domain overfitting. Further analysis of cross-domain or cross-model transferability would strengthen claims of general utility.

**Questions:**

If a probe trained on one judge is applied to another, how does performance degrade? This would test whether latent representations capture universal judgment signals.

---

> ### Author Response · Authors · 2025-12-02
> **Author Response to Reviewer bhBs (1/2)**
>
> We thank the reviewer for the positive review with great suggestions. We are pleases the reviewer finds our paper “Well-motivated and methodologically sound”, featuring “comprehensive benchmarking” and providing “Insightful empirical findings”. Below, we respond to the individual concerns, suggestions, and questions mentioned in the review.
>
> ---
>
> > **(W1)** Novelty. While the latent probing approach is interesting, the “verifier-style” probability concept has been explored previously in domains like math/code.
>
> In our revision, we have clarified the contribution. Specifically, as the reviewer notes, this approach is common in mathematical and coding settings, where an objective ground truth exists. In these contexts, models are trained to predict whether a given response is correct. Our innovation transfers this setup to general-purpose prompts, where the notion of “correctness” does not exist. Inspired by encouraging results from evaluating prompt-image correspondence in synthetic images [1], we use base LLMs to judge whether a response is good ($P(\text{yes}) = 1$). Unlike typical verifier models, these base LLMs are not fine-tuned. We have added this in Sec. 3.2 (below Eq. (2)).
>
> [1] Lin et al.: Evaluating text-to-visual generation with image-to-text generation. In ECCV, 2025.
>
> ---
>
> > **(W2)** Distinguishing LLM-as-a-Judge vs. Reward Models. The motivation would benefit from a clearer conceptual separation between these paradigms. Including a SOTA reward model (e.g., top of RewardBench) (or justification for the choice of already-included models that are fine-tuned for judging) in the determinism and calibration analysis could reveal whether reward-model scores are inherently more deterministic.
>
> Thank you for the great suggestion. Indeed, trained reward models share similarities with our approaches, but there are also important differences. The main shared aspects are that responses are evaluated as a scalar score instead of a discrete rubric, and that training is used to create reward models (we also train our latent probes). Regarding training, the two approaches massively differ in scale: While we utilize a few thousand response pairs, Skywork-Reward-V2, for example, trains on 26 million automatically rated and human rated preference pairs. Also, reward modeling approaches train the full model and not only an external probe, like us. This has important consequences: A trained reward model can no longer be queried for rationales, and might lose reasoning capabilities necessary to judging hard cases. Reward models can also lose prompt robustness, while leaving base LLMs unchanged allows for prompting with arbitrary instructions and rubrics. As our results show, advantages of specifically trained LLM-as-a-judge models (Selene, Prometheus) over general-purpose LLMs are very small, so the prompt generalization is a valuable property of our setup.
>
> Regarding determinism, we acknowledge that reward models are also deterministic. Regarding calibration, we observe that scores of reward models also concentrate in a narrow range (when scaled on the same spectrum like other models), and this range is narrower than probability-weighted ratings, for example. This is in Fig. 7. We have added this analysis in Appendix A.3 in our revision, and we have clarified the comparison to reward models in our related work section (Sec. 2, “Reward Models”).
>
> ---
>
> > **(W3)** Comparative baseline. Table 1 currently benchmarks against GPT-4 judgments but not against post-trained reward models. Comparing agreement with a strong, fine-tuned reward model would add to the evaluation.
>
> We have followed the reviewer’s suggestion and added results when using Skywork-Reward-V2 [1] to evaluate pairwise preference in the Tülu dataset (Table 1). This model is a state-of-the-art reward model, currently ranking #1 on the Rewardbench Leaderboard [2], as suggested by the Reviewer. We note that overall, agreement rises slightly by 1 to 2\%, but observations remain the same.
>
> [1] Liu et al.: Skywork-Reward-V2: Scaling Preference Data Curation via Human-AI Synergy. In arxiv, 2025.\
> [2] https://huggingface.co/spaces/allenai/reward-bench
>
> ---
>
> > **(W4)** Sensitivity to Temperature. The determinism experiment should test multiple temperature settings to confirm that observed variance is not an artifact of decoding randomness.
>
> Thank you for this suggestion. We have added an experiment where we ablate the effects of temperature on our observations. In the revision, this experiment is in Appendix A.1 and we have added a reference in Sec. 3.1. In summary, we find that while a lower temperature, i.e. more deterministic outputs, gives higher stability of scores (which is expected), for all tested temperatures (0.1, 0.2, 0.4, 0.6, 0.8, 1.0) there is substantial non-determinism, which confirms our observation of excess epistemic uncertainty of judge LLMs regarding the correct rating.

---

> > ### Author Response · Authors · 2025-12-02
> > **Author Response to Reviewer bhBs (2/2)**
> >
> > ---
> >
> > > **(W5)** Explanation of why instability occurs. The paper attributes inconsistency and poor calibration to discrete categories and stochastic decoding but does not elaborate. A dedicated section explaining the intuition behind these mechanisms would round out the storyline.
> >
> > In our revision, we have included a more detailed discussion in Sec. 3.1 (“Discussion”). In summary, we think the observed instability is caused by the judge model's epistemic uncertainty when constrained to discrete scoring rubrics, which results as increased entropy during stochastic decoding. At the same time, calibration issues are likely due to bias toward linguistic fluency, i.e. any fluent and relevant answer would already achieve a decently high score. However, this is not useful given the general capability of modern LLMs to produce fluent and relevant responses.
> >
> > ---
> >
> > > **(W6)** Potential label noise. It would be interesting to replicate Table 4 using only clearly separable examples (e.g., where human preference strongly agrees across annotators) to test robustness (Frequent ties might indicate noise or low-signal regions in human-labeled preference data.)
> >
> > We have checked all 10 benchmarks evaluated in Table 2, and found only PreferenceBench and UltraFeedback provide finegrained ratings that allow us to distinguish clear cases. In both cases, scores are on a scale from 1-5 and we only keep pairs where the rating difference between the preferred and the dispreferred answer is larger than 2. Results are in Appendix Appendix A.3 (“Evaluating Pairwise Benchmarks on Clearly Separable Response Pairs”). We find that these cleaned benchmarks are too easy and all methods (with few outliers) solve them almost perfectly. Therefore, they are not suitable to assess performance differences of the different methods.
> >
> > ---
> >
> > > **(W7)** Please add confidence intervals for the mean estimates in Table 4.
> >
> > In our revision, we have added confidence intervals for all benchmark accuracies in Table 2 (numbering has changed in the revision), as the reviewer suggested. We calculate the 95\%-CIs for the accuracies using the Clopper-Pearson exact interval, as accuracies are equivalent to the maximum likelihood estimate of a Bernoulli distribution. Because showing all the confidence intervals takes up a significant amount of space, we replicated Tab. 2 including the CIs as an additional table in the Appendix (Tab. 8) and kept the table in the main paper unchanged in favor of readability.
> >
> > ---
> >
> > > **(W8)** Generalizability of Latent Probes. The finding that probes trained on Tülu responses perform best raises concerns about domain overfitting. Further analysis of cross-domain or cross-model transferability would strengthen claims of general utility.
> >
> > Thank you for raising this point. In our current results (Table 2 and 3), probes trained on the Tülu Preference Mixture generalize well across a wide range of benchmarks (10 different pairwise benchmarks and 7 single-rating benchmarks), and we do not observe clear signs of domain overfitting. In our revision, we have now also included an analysis of the unsupervised data used to train probes (Appendix B, “Data”). We find that mixes of several models yield good performance, however probes can be sensitive to this, and in only one case (GLM4-9B versus Qwen-4B), performance degrades substantially. This shows while many settings work, the choice of models from which to take preferred and dispreferred responses matters.
> >
> > ---
> >
> > > **(Q1)** If a probe trained on one judge is applied to another, how does performance degrade? This would test whether latent representations capture universal judgment signals.
> >
> > Unfortunately, we are unable to evaluate this idea because the latent spaces of different LLMs have different dimensionality. Therefore, a probe trained on one LLM cannot apply to another. However, we agree that this is a very interesting direction. Future work could investigate aligning activations from different LLMs in a joint space or concatenating activations from multiple LLMs as an ensemble. This is costly in practice, however, as it requires running multiple LLMs to judge a single response.
> >
> > ---
> >
> > Thank you very much once again for the detailed review. The suggestions have helped us improve the paper, and we have addressed all the mentioned points in our revised version.

---

### Official Review · Reviewer_9WLH · 2025-10-31

**Soundness:** 2
**Presentation:** 3
**Contribution:** 2
**Rating:** 4
**Confidence:** 5

**Summary:**

This paper addresses single-response, reference-free evaluation in LLM-as-a-Judge settings, identifying two systematic issues: (1) sampling instability, and (2) poor calibration leading to score compression and ties. To address these, the authors propose Latent Judges deriving ratings from internal model signals: (1) probability-weighted scores via next-token distributions, (2) verifier-style binary probabilities, and (3) latent probes using linear classifiers on hidden activations. Evaluations across more than 10 benchmarks show these approaches match or surpass standard prompting, particularly for pairwise accuracy and listwise ranking in Best-of-N selection.

**Strengths:**

1. Well-Motivated Problem
The paper clearly articulates why single-response, reference-free evaluation matters for practical applications (Best-of-N, GRPO, multi-teacher distillation, routing). The systematic analysis in Section 3.1 convincingly demonstrates the limitations of current approaches:
Even the best models achieve only 70-80% agreement with mode across 10 samples (Figure 1)
Severe score compression near the top of the scale (Figure 2)
Large gaps between strict and lenient agreement metrics (Table 1)

2. Comprehensive Experimental Evaluation
The experimental design is thorough:
12 judge models including specialized judges (Prometheus, Selene)
10+ diverse benchmarks covering both pairwise and single-rating settings
Practical applications (listwise ranking, LLM routing)
Multiple prompt templates and probe architectures tested

3. Diversity of methodology
The three proposed approaches are complementary and theoretically grounded:
Probability-weighted ratings directly leverage model uncertainty
Verifier-style extends successful approaches from verifiable domains
Latent probes can recover signals when output logits are miscalibrated
The connection to KTO loss (Appendix D) is elegant and adds theoretical depth

4. Empirical Gains
Results show consistent improvements:
Pairwise benchmarks: Up to 5 percentage points improvement in average accuracy (Table 4)
Listwise ranking: Superiority in Spearman correlation with GPT-5 rankings (Table 6)

**Weaknesses:**

1. Training Instability - Acknowledged but Not Solved
In Section 4.4, lines 369-375, the authors acknowledge the problem: "we notice considerable variation over different training runs." This is serious - it means training the probe multiple times with the same data and settings produces different performance each time.
But the paper doesn't quantify this problem. What does "considerable" mean? Is the standard deviation 0.02 or 0.10? What's the gap between best and worst performance - 5% or 20%? Did they train 10 times or 100 times? None of this basic information is provided.
Imagine a practical scenario: I'm deploying a probe at my company. The first training run achieves 75% performance. Looks good. But my colleague says "maybe you got lucky, try again." The second run gives 70%. "Hmm, which is real?" The third gives 78%.


2. Data Requirements - Hidden costs and circular logic
Probes require labeled preference data. The paper presents two approaches: using existing datasets like Tulu, or generating data with strong/weak model pairs. There's also circular logic here. We're trying to build a good evaluator, but to do so we already need a good evaluator (the strong model).


3. Single-Rating Performance: Severe Degradation Compared to Alternatives
The paper's title is "Reference-Free Rating of LLM Responses." "Rating" - evaluating individual responses - is the core task, yet probes perform worst at this. They're okay for pairwise comparison but worst for single rating.


4. Cost-Benefit and Risks in Practical Implications

**Questions:**

1. Training Instability - Acknowledged but Not Solved
How many times should a researcher or user train? Can I pick the best run? Or should I average them? The paper provides no guidance for these practical questions. Moreover, there's no proposed solution. The paper only says "more distinct training data helps mitigate this," but doesn't specify how much it helps or exactly how to prepare such data.

2. Data Requirements - Hidden costs and circular logic
For labelling preference data,
The first approach : How much hidden cost we need?
The second approach : We need to choose strong and weak models, but how different should they be? The paper only says "more clearly separated models make training more stable." Do you need a huge gap like GPT-4 vs Llama-3B? Would GPT-4 vs Llama-70B suffice? There are no concrete criteria. If we're using GPT-4 to generate training data, why not just use GPT-4 as the evaluator? Why bother training a probe?

3. Single-Rating Performance: Severe Degradation Compared to Alternatives
3-1. In Table 5, for single-rating benchmarks, probes achieve Pearson correlations averaging 0.40-0.48. This is much lower than probability-weighted methods at 0.59-0.64 - almost a 20% difference. The authors explain why in lines 289-293: the BCE (Binary Cross-Entropy) objective function "squashes outputs toward extremes." This means probes tend to output values close to 0 or 1, and struggle to predict intermediate values.

This is exactly the same problem as the verifier-style method. The verifier also has P(yes) concentrated at 0 or 1, reducing discriminative power. If probes suffer from the same extreme value bias as verifier-style methods, what advantage do they offer to justify the additional training cost?

3-2. "Rating" - evaluating individual responses - is the core task, yet probes perform worst at this. They're okay for pairwise comparison but worst for single rating. Why must we use BCE? Could we check other loss functions like MSE (Mean Squared Error)? What about focal loss or other variants?

4. Missing Cost-Benefit Analysis
Using probes requires: Preparing training data (time, cost), Multiple training attempts, Selecting the best probe,
Extracting hidden states during inference. Is all this cost worthwhile? How could we measure Cost-Benefit ?

---

> ### Author Response · Authors · 2025-12-02
> **Author Response to Reviewer 9WLH (1/2)**
>
> We thank the reviewer for the detailed review and suggestions to improve our paper and clarify our contributions. We appreciate that the reviewer highlights that we investigate a “well-motivated problem” and provide a “comprehensive experimental evaluation” covering “10+ diverse benchmarks”. The reviewer also acknowledges the “diversity of methodologies” that we evaluate, which yield “consistent improvements”. Below, we reply to the mentioned concerns individually and describe how we address them in the updated version.
>
> ---
>
> > **(W1)** The authors acknowledge the problem: "we notice considerable variation over different training runs." This is serious - it means training the probe multiple times with the same data and settings produces different performance each time.
>
> We apologize for the confusion. We did not mean that with the same data and settings, results will be different. In fact, if these are kept the same, the training is deterministic (given a fixed random seed). We noticed, however, that the probes tend to be sensitive to some hyperparameters and especially the data. However, we acknowledge that our statement may have been imprecise and we have revised it in the revision. We investigated hyperparameter sensitivity more closely and noticed that the main factors that lead to the perceived sensitivity are (i) the layer index from where we extract activations; and (ii) the data we use to train the probe (to some extent).
>
> We discuss data below in a separate answer. Regarding hyperparameters and layer extraction index, in our revision, we have included an extensive evaluation on Phi-4 in Appendix B (“Hyperparameter Study”). There, we show that the probe’s performance is robust to a range of batch sizes and regularization strengths on the score’s magnitude. The choice of layer to extract activations from has a significant impact, though, as we show there. In particular, it is necessary to extract activations from intermediate layers instead of final layers (close to the language modeling head).
>
> ---
>
> > **(Q1)** How many times should a researcher or user train? Can I pick the best run? Or should I average them?
>
> Thank you for asking these questions. Essentially, this is a problem of hyperparameter selection. After extracting and caching activations from judge models, we train multiple probes with different parameter settings and select the best performing model, using a validation set. Furthermore, improvements to the setting, such as the data can improve performance, we discuss this below.
>
> ---
>
> > **(W2/Q2)** The paper presents two approaches: using existing datasets like Tulu, or generating data with strong/weak model pairs. There's also circular logic here. We're trying to build a good evaluator, but to do so we already need a good evaluator (the strong model). The first approach : How much hidden cost we need? The second approach : We need to choose strong and weak models, but how different should they be? The paper only says "more clearly separated models make training more stable." Do you need a huge gap like GPT-4 vs Llama-3B? Would GPT-4 vs Llama-70B suffice? There are no concrete criteria. If we're using GPT-4 to generate training data, why not just use GPT-4 as the evaluator? Why bother training a probe?
>
> We are happy to clarify this question. In our paper, we use two setups: (1) Training the probe on publicly available GPT-4 rated data, i.e., the Tülu Preference Mixture; (2) Training on responses from pairs consisting of a stronger and a weaker model. Here, our aim is to evaluate how big the gap is between “ground-truth” GPT-rated data (which, as the reviewer mentions, requires additional cost to create) and unsupervised data from model pairs. Option (1) is meant as an upper bound, however we would still like to stress that we only used openly available, i.e. already existing, data without generating new preference pairs from GPT judgements. Here, we want to stress that the performance gap between option (1) and option (2) is very small in Table 4 and 5, meaning that we can replace GPT-rated pairs with unsupervised pairs with little or no loss in performance.
>
> For option (2), we observe that indeed the selected model pairs have influence on the performance. We evaluated several combinations of models and included this evaluation in the revision in Appendix B (“Data”, Fig. 11). There, we show that several data mixes yield comparable performance and only one evaluated setting deviates. This shows that care must be taken when selecting models, but already reasonable comparisons, for example EXAONE-3.5-7.8B vs. Llama3 3B closely follows the performance of GPT-rated pairs. This also answers the question whether we need huge gaps: this is not the case. However, larger and better models are expected to improve performance further.

---

> ### Author Response · Authors · 2025-12-02
> **Author Response to Reviewer 9WLH (2/2)**
>
> ---
>
> > **(W3)** The paper's title is "Reference-Free Rating of LLM Responses." "Rating" - evaluating individual responses - is the core task, yet probes perform worst at this. They're okay for pairwise comparison but worst for single rating.
>
> We agree that performance of probes on single-rating tasks falls behind other models. However, we would like to stress that we evaluate three methods that use latent information, i.e. verifier-style ratings, probability-weighted scores, and probes. Also on single-rating tasks, the probability-weighted method outperforms traditional LLM-as-a-judge (Table 3), while the setup of the benchmarks is tailored towards discrete scores and thus favors the baseline. Our results show that on every evaluated task, scores based on latent information (mostly probability-weighted or probes) outperform the baseline, i.e. traditional LLM-as-a-judge scoring. In our revision, we have added a new section, i.e. Sec. 6, where we discuss and clarify these aspects.
>
> ---
>
> > **(Q3-2)** Why must we use BCE? Could we check other loss functions like MSE (Mean Squared Error)? What about focal loss or other variants?
>
> Thank you for this great suggestion. We have now evaluated a range of different losses, in particular MSE, Focal Loss, Huber Loss, Hinge Loss, BCE with polynomial extension ($\ell_{\text{BCE}) + \epsilon(1 - p_t)$ to focus on hard samples). The results are in Appendix B (“Training”) in the revision, specifically in Eq. 5-9 and Fig. 9. We find that BCE and BCE with polynomial extension perform best and are most stable across different batch sizes. These results justify our choice of BCE. MSE also performs well for larger batch sizes, but is unstable for lower batch sizes. Performance of Focal Loss falls behind other formulations.
>
> ---
>
> > **(Q3-1)** the BCE (Binary Cross-Entropy) objective function "squashes outputs toward extremes." This means probes tend to output values close to 0 or 1, and struggle to predict intermediate values. This is exactly the same problem as the verifier-style method. The verifier also has P(yes) concentrated at 0 or 1, reducing discriminative power. If probes suffer from the same extreme value bias as verifier-style methods, what advantage do they offer to justify the additional training cost?
>
> In the revision (Appendix B, “Training”, Fig. 10) we now also evaluated the effect of different losses on the resulting score distributions. There we show that different loss functions yield distinct distributions. For example, MSE and Focal Loss produce approximately Gaussian score distributions, whereas BCE and its polynomial extension appear more skewed. However, the polynomial extension covers the full score spectrum more effectively. Finally, Huber loss yields a bimodal distribution but generally resembles the BCE with polynomial extension. These results demonstrate that the loss function has a significant impact on the score distribution. However, more balanced distributions do not necessarily lead to improved benchmark results.
>
> The distinct advantage of latent probes versus verifier-style ratings is that probes also work when the LLM does not follow the prompts and does not return useful or calibrated logits. We find this for Prometheus-7B, which is essentially unusable for verifier-style ratings (see Table 2 and 3). Naturally, training probes enable much better control over the resulting score distribution. We have added a clarifying statement in Sec. 3.2 (“latent probing”).
>
> ---
>
> > **(W4/Q4)** Missing Cost-Benefit Analysis Using probes requires: Preparing training data (time, cost), Multiple training attempts, Selecting the best probe, Extracting hidden states during inference. Is all this cost worthwhile? How could we measure Cost-Benefit?
>
> Thank you for raising this issue. First, we would like to notice that verifier-style ratings and probability-weighted scores do not incur any additional overhead compared to standard inference. Probe training indeed has a certain cost associated, however, we think this can be justified: Data preparation requires inference from LLMs, unless we use available data, such as Tülu. Training linear probes, once activations are extracted and caches, is computationally very cheap. Finally, at inference time, latent probes can actually reduce computational requirements. For example, if we apply the linear probe on activations of layer 31 of 40 (as we do for Phi-4), we do not need to evaluate the subsequent layers, which could save around 25\% of the full forward pass. In our revision, we have included these considerations in the new Sec. 6 (“Discussion and Limitations”).
>
> ---
>
> We would like to thank the reviewer again for the great questions and constructive feedback. In our revision, we have included corresponding clarifications and additional ablations, which we believe fully address the reviewer's concerns.

---

### Official Review · Reviewer_urQp · 2025-11-05

**Soundness:** 2
**Presentation:** 2
**Contribution:** 2
**Rating:** 2
**Confidence:** 3

**Summary:**

The paper studies the problem of using an LLM as a judge to rate LLM responses in the non-verifiable setting, typically the text format responses are rated by the judge LLM on the scale of 1-5 or 1-10. These ratings may have large variability under sampling, poor calibration and frequent ties leading to suboptimal input for downstream tasks. The authors evaluate 12 judge-LLMs on a prompt + chosen/rejected response dataset to show that by and large the LLM judges are inconsistent in their outputs under sampling, and poorly calibrated with the chosen responses.
To mitigate this, the authors propose three techniques to derive response scores from latent knowledge: probability-weighted ratings, verifier-style ratings and latent probing. The first method gives a deterministic score which is the probability weighted rating, the second one is the case when only yes/no answers are elicited and the rating is the probability of yes, while the last method trains a linear regressor over the activations at the next token to predict a quality score.
The authors conduct experiments using several pairwise response preference and single rating benchmarks, with GPT-4 and human rates as the reference, for 6 judge LLM models, evaluating the proposed rating methods against the 10-scale rating as the baseline. The experiments empirically show some performance improvements.

**Strengths:**

The methods are novel and aim to address important challenges faced in using LLM as judge, which is required for several important downstream applications.

**Weaknesses:**

1. The experiments presented in the paper only show modest accuracy improvements over the 10-point baseline. Further, no one method seems to be consistently the best across the judge-LLMs or the benchmarks.
2. The experiments only measure the quality metrics against reference human or GPT scores. However, the methods are not evaluated on their impact on the performance of downstream tasks like ranking, distillation or RL policy optimization.
3. The paper does not provide much analytical insights on why the proposed techniques would improve performance.

**Questions:**

1. The experiments are mostly on smaller LLMs as judges .. can the authors discuss whether/why they expect their techniques to apply to larger LLMs?
2. How is the temperature tuned for the probability-weighted rating and verifier-style rating methods?
3. What is layer $l$ on line 214?
4. Why is the variance of latent probe not included in Fig. 3?

---

> ### Author Response · Authors · 2025-12-02
> **Author Response to Reviewer urQp (1/2)**
>
> We thank the reviewer for the thoughtful review, highlighting our “methods are novel” and “address important challenges” for “several important downstream applications”. Below, we discuss the concerns mentioned by the reviewer in detail.
>
> ---
>
> > **(W1)** The experiments presented in the paper only show modest accuracy improvements over the 10-point baseline. Further, no one method seems to be consistently the best across the judge-LLMs or the benchmarks.
>
> We agree that while different methods yield best performance on the different tasks, our analysis clearly shows that LLM judges operating on latent information are superior to traditional Likert-style metrics. In Tables 2, 3, and 4, it is generally probability-weighted ratings or latent probes that perform best. This supports our argument that latent information provides a better basis to rate responses than text outputs of LLM judges.
>
> Additionally, two of the three evaluated latent judge methods (verifier-style judge and probability-weighted) are training-free, so comparing them is cheap, and training probes only require extracting activations for a few thousand paired samples, i.e. training is also lightweight. Our work shows that the traditional LLM-as-a-judge paradigm can be improved effectively without any, or with very light, additional training cost. This contrasts to dedicated reward models that require millions of training samples.
>
> In our revision, we have added a section (Sec. 6) discussing the performance and efficiency trade-offs, which we hope addresses the Reviewer’s concerns.
>
> ---
>
> > **(W2)** The experiments only measure the quality metrics against reference human or GPT scores. However, the methods are not evaluated on their impact on the performance of downstream tasks like ranking, distillation or RL policy optimization.
>
> In our paper, we already include two downstream evaluations: In Sec. 5.1, we evaluate ranking against labels derived from GPT-5 (reasoning) showing that indeed, probability-weighted scores and latent probes outperform traditional LLM-as-a-judge scoring schemas. Furthermore, in Sec. 5.2, we show how Phi-4 derived scores (probability-weighted) can help understand the predictability of LLM routing decisions from prompt semantics. In our revision, we have clarified the purpose of Sec. 5 in an introductory paragraph.
>
> While we already evaluate ranking as the reviewer suggested, distillation and RL policy optimization are exciting directions for future research. Because of the significant contributions over existing work, these would constitute independent contributions and are hence out of scope of this paper.
>
> ---
>
> > **(W3)** The paper does not provide much analytical insights on why the proposed techniques would improve performance.
>
> Thank you for the great suggestion. We have added a discussion in Sec. 6 of the revision to address specific reasons why latent information helps assess responses better:  We argue that it is inherently more fine-grained. Sampling from the judge LLM to obtain a rating necessarily loses information. This process simplifies the rich response representation that the judge builds internally into a single number. Logits and internal activations encode richer features that distinguish between responses. These include stylistic, semantic, or discursive features, as well as aspects related to factuality and helpfulness. More direct access to these attributes, even in aggregated form as a single scalar value, facilitates more reliable response assessment.

---

> > ### Author Response · Authors · 2025-12-02
> > **Author Response to Reviewer urQp (2/2)**
> >
> > ---
> >
> > > **(Q1)** The experiments are mostly on smaller LLMs as judges .. can the authors discuss whether/why they expect their techniques to apply to larger LLMs?
> >
> > We are happy to clarify this. We evaluate LLMs of different sizes, up to 70B parameters (LLama 3.3, Selene-1, Qwen2.5). While large models (70B) are overall stronger than smaller models (e.g. Qwen3 14B, Phi-4 (14B)), this advantage is limited. For example, Phi-4 performs almost on par with models 5x its size. This shows that model quality is more important in our setting than model size, and small models may in practice suffice to provide useful ratings. In the revision, we have clarified the range of our models in Sec. 3.1 (“Data and Models”) and in Sec. 4.3 (first paragraph). We would also like to note that all our methods are not dependent on the size of the LLM and performance is expected to increase further with larger models.
> >
> > ---
> >
> > > **(Q2)** How is the temperature tuned for the probability-weighted rating and verifier-style rating methods?
> >
> > In the paper, we always used temperature 1 as a natural choice and did not tune the temperature. Following the suggestion by the reviewer, we have evaluated the temperature in a range from 0 to 2. The result is in the revision, in Appendix A.3 (“Ablating the Temperature for Verifier-Style and Probability-Weighted Scores”). There, we show that higher temperature yields better results for verifier-style ratings (possibly related to the improvements we get with calibration, please see our response to reviewer XxtF). Probability-weighted scores are robust to changes in temperature.
> >
> > ---
> >
> > > **(Q3)** What is layer l on line 214?
> >
> > With the layer $l$, we are referring to a specific layer of decoder-only LLMs, i.e. the respective index of the transformer block after which we extract representations. In our experiments, we use layers according to the size of the evaluated LLMs: For Qwen3-14B and Phi-4, which have both 40 layers, we use layer 31, and for Prometheus-7B, which is smaller (32 layers), we use layer 21. We clarified this in Sec. 3.2. These layers were selected based on their superior performance, and we think this provides an interesting insight that intermediate layers may encode richer information about the input than final layers. In our revision (Appendix B, “Hyperparameter Study”, Fig. 12), we added a systematic study on Phi-4 showing that indeed intermediate layers perform significantly better than final layers for probe training.
> >
> > ---
> >
> > > **(Q3)** Why is the variance of latent probe not included in Fig. 3?
> >
> > We have now included an ablation of the score distribution for latent probes in Appendix B (“Training”, Fig. 10). There, we show how different losses affect the score distribution of linear probes trained on Phi-4 embeddings. We can see that different losses lead to different distributions, some less skewed than others, but we do not observe that less skew results in better performance.
> >
> > ---
> >
> > We thank the reviewer once again for the helpful suggestions and constructive feedback, which have helped us significantly improve our paper. We believe that our revision and responses fully address the reviewer's concerns.

---

### Author Response · Authors · 2025-12-02
**General Response**

Dear AC and Reviewers,

Thank you very much for your time and efforts at ICLR 2026. We appreciate the insightful reviews. We have now uploaded our revised version and responses to all reviewers. The revised version includes 9 pages of new content mostly in the supplementary, increasing the original length by approx. 43\%.

---

Our submission received scores 6,6,4,2, which demonstrates that our work is of interest to the community. All reviewers expressed that we present novel methods that impact important applications. Reviewers also highlighted that we motivate our paper well (9WLH, bhBs, XxtF) and our results are methodologically sound (bhBs, XxtF). In particular, reviewers mentioned the comprehensive evaluation on a large number of evaluated benchmarks (9WLH, bhBs, XxtF) and our results show empirical gains (9WLH) that have significance for several ML communities (XxtF).

---

Our main contributions are threefold: We first empirically show that the standard LLM-as-a-judge paradigm is limited and suffers from miscalibration and instability (scores are inconsistent under repeated sampling and concentrate on a narrow range of the scale). We then describe and evaluate three methods using logits or internal activations that mitigate these problems and show improved performance on 17 (in total) standard LLM-as-a-judge benchmarks. Finally, we show how these methods also improve and benefit downstream tasks, in particular listwise response ranking and model routing.

---

Reviewers suggested interesting and important ablations as well as more detailed and nuanced discussion of our findings. We have provided these in the revision (marked in blue), and we clarified several aspects mentioned by the reviewers throughout the paper. In our overview of changes below, we summarize how we revised the paper based on the concerns of the reviewers and state precisely where in the paper we have addressed them. In our individual responses to each reviewer, we answer the mentioned concerns in detail.

---

> ### Author Response · Authors · 2025-12-02
> **Overview of Changes in the Rebuttal Revision**
>
> ---
>
> > Clarification regarding novelty (bhBs: W1; XxtF: W1)
>
> We have added an explanation of the novel and paradigm-shifting aspects of our work to Sec. 2 (“Latent Probing”): While latent information is used for other tasks as detailed above, we
> systematically evaluate it for rating responses. This changes the evaluation paradigm from ratings as text outputs of the judge LLM to the use of the rich information judge LLMs encode internally.
>
> ---
>
> > Discussion and Limitations: No single method performs best (urQp: W1; 9WLH: W3; XxtF: W1) and cost considerations (9WLH: W4/Q4)
>
> We have added a new section (Sec. 6), where we discuss these aspects. We highlight that methods based on latent information always perform better than the standard LLM-as-a-judge paradigm and that latent probes can in practice save cost by not evaluating the final layers of LLM (after the layer from which we extract activations).
>
> ---
>
> > Downstream Applications (urQp: W2)
>
> We have added a paragraph at the start of Sec. 5 (line 450) where we clarify that we indeed evaluate on downstream tasks, especially listwise ranking, as reviewer urQp suggested.
>
> ---
>
> > Data requirements of latent probes (9WLH: W2/Q2; XxtF: W3)
>
> We have clarified an imprecise statement in Sec. 4.4 (line 431), and we now ablate the data requirements in Appendix B (“Data”, Fig. 11).
>
> ---
>
> > Using activations from intermediate layers for latent probe training (urQp: Q4; XxtF: Q3)
>
> Sec. 3.2 (line 246) now states explicitly that we train probes on activations from intermediate layers, and we ablate the choice of layer in Appendix B (“Hyperparameter Study”, Fig. 12).
>
> ---
>
> > Relation to Reward Models (bhBs: W2 + W3)
>
> We have added a discussion on reward models in Sec. 2 (“Reward Models”) and added a comparison to reward model ratings in Tab. 1. Finally, we assess calibration of reward models in App. A.4 and Fig. 7.
>
> ---
>
> > Relation of verifier-style ratings to verifiers in math/coding (bhBS: W1)
>
> The differences of our verifiers-style ratings to trained verifiers in settings where objective ground truth exists, like math or coding, are now discussed in more detail in Sec. 3.2 (line 234).
>
> ---
>
> > Calibration of verifier-style ratings (XxtF: Q2)
>
> In Tab. 3, we added results where we scale verifier-style ratings using a small validation set. The method is explained in Sec. 4.4 (“Calibration of Verifier-Style Scores”). We observe calibration improves performance significantly.
>
> ---
>
> > Model Sizes and Evaluation of Large Scale Models (urQp: Q1)
>
> We added a clarification about the model sizes (up to 70B) in Sec 3.1 (line 146) and in Sec. 4.3 (line 295).
>
> ---
>
> > Effect of temperature on instability of ratings (bhBs: W4)
>
> We added the requested ablation of temperature in Appendix A.1 and referenced this in Sec. 3.1 (line 151).
>
> ---
>
> > Additional discussion of instability and insufficient calibration of LLM-as-a-judge (bhBS: W5; urQp: W3)
>
> Sec. 3.1 now includes an additional paragraph (“Discussion of Findings”) where we discuss results on weaknesses of the standard LLM-as-a-judge method and analyze the causes.
>
> ---
>
> > More additions (in the supplementary material)
>
> * We replicated Table 2 with 95% confidence intervals (Clopper-Pearson exact intervals), as requested by reviewer bhBs (W7)
> * We evaluated pairwise benchmarks on clearly separable instances (PreferenceBench and UltraFeedback, where such information is available. Results are in Appendix A.3 (line 1101, Tab. 9). This responds to reviewer bhBs (W6)
> * In Appendix A.3 (line 1134, Fig. 6) we show how probability-weighted ratings and verifier-style ratings can be improved by tuning the temperature. This responds to reviewer urQp (Q2)
> * In Appendix B (line 1252), we ablate different losses for latent probes and find that BCE (and variants) perform best. This responds to reviewer 9WLH (Q3)
> * We report further hyperparameter ablations for latent probe training in Appendix B (“Hyperparameter Study, line 1362) in response to reviewer 9WLH (W1/Q1)

---

### Meta-Review · Area_Chair_SF4F · 2026-01-06

**Summary:**

The paper presents a family of "latent information" approaches to the LLM-as-a-judge problem. These contrast to conventional decoding approaches, which are shown to suffer empirical and conceptual limitations. Reviewers found the work to be interesting, while raising some concerns:

- **Limited novelty**. Multiple reviewers noted that the core ideas of the paper are of limited technical novelty, albeit well executed in the present context.

- **Lack of consistent empirical trend**. Multiple reviewers noted that there is no clear winner amongst the various latent approach methods. Further, the gain of latent methods on single-rating correlation tasks was seen as marginal.

- **Generalizability of latent probes**. Multiple reviewers expressed concern that the efficacy of the latent probe method depends on the training data source, raising doubts of overfitting, and qualification to the claim of a generic off-the-shelf solution.

- **Training instability**. One reviewer expressed concern that the results with latent probes exhibit significant variation across runs.

- **Role of reward models**. One reviewer noted that there are strong conceptual similarities to reward models, which deserve more discussion and possibly experimentation.

- **Lack of downstream evaluation**. One reviewer noted that the downstream gains of the proposed method are not clearly demonstrated. Rather, gains are shown only in terms of alignment to the target correctness scores (which are indicative but not conclusive).

- **Lack of analytical insights**. One reviewer noted that the paper does not provide much in the way of analytical insights as to _why_ latent methods outperform traditional ones.

**Reviewer Concerns:**

- **Limited novelty**. The authors acknowledged that similar ideas have appeared in different contexts. They argued that the application to the LLM-as-a-judge setting, and handling settings where there is no objective ground truth, are the major contributions. They also updated the discussion of the contributions and related work.
  - *Partially addressed*. There seems to be agreement that the novelty is in the application of existing ideas, with significant implications. This is certainly reasonable as a contribution. However, this point requires further qualification (see below).
  - *AC's comment*. The authors' treatment of related work in the LLM-as-a-Judge literature appears incomplete.
    - Please see, e.g., Chiang et al., "TRACT: Regression-Aware Fine-tuning Meets Chain-of-Thought Reasoning for LLM-as-a-Judge" and references therein for a very similar methodology to the "probability-weighted ratings" approach. The paper and references therein discuss both zero-shot and fine-tuned models.
    - Outside of the LLM-as-a-judge setting, more relevant works exist; e.g., Zausinger et al., _Regress, Don’t Guess – A Regression-like Loss on Number Tokens for Language Models_.
    - While certainly there are novel aspects of the present work -- such as considering alternate latent approaches, and demonstrating the limitations of classical decoding approaches) -- such related work requires discussion, and certainly suggests tempering of claims of introducing a paradigm shift.

- **Lack of consistent empirical trend**. The authors emphasized that a consistent trend holds wherein latent methods outperform traditional ones, even if the _specific_ winning latent method can vary. The authors noted that for the single-rating correlation tasks, the scores exhibit extreme bi-modality. They found by performing calibration, the latent methods can perform much better (updated Table 3).
  - *Mostly addressed*. We tend to agree that the main point, that latent methods are superior to traditional ones, is generally supported by the results. We do agree with the reviewers that it is slightly unsatisfying that there is no clear understanding of why a particular latent method should be preferred.

- **Generalizability of latent probes**. The authors clarified that the results on the Tulu mixture represent an upper bound, while the results on strong-weak pairs represent what is to be practically considered. They also noted that the gap between these methods is small in Table 4 and 5. Finally, the authors added more details in Appendix B of the sensitivity of the latter "unsupervised" approach to the specific choice of model pairs, generally finding them to be robust.
  - *Mostly addressed*. The author's response makes sense, but the latest revision still has scope to be much clearer on this point.

- **Training instability**. The authors clarified that the variation does not happen with the same data and setting, but rather that the results vary with changes to hyper-parameters such as the layer index for extracting the probe, and the data.
  - *Mostly addressed*. Further clarification in the text is advised.

- **Role of reward models**. The authors acknowledged the conceptual similarity to reward models. However, they noted distinctions such as the fine-tuning of all parameters in reward models (rather than a separate probe), and the typical usage of millions of training examples. The authors also presented new results in Table 1, where the Skywork reward model is used to assess judge model outputs. This was found to yield largely similar results as with a GPT-4 judge model.
  - *Partially addressed*. The added discussion of _conventional_ reward models is a useful addition. There is however scope for further discussion of _generative_ reward models, and their relation to the present work.
  - *AC's comment*. It is not clear whether some of the points are immovable requirements for reward models, rather than being conventional practice. For example, is the claim that it is not at all feasible to have a generative model that has an auxiliary head that models rewards? There is much recent work that suggests this is not the case; e.g., Mahan et al., Generative Reward Models. Regarding reward models typically requiring millions of examples, what is the conceptual reason that the present approach can get away with far fewer examples?

- **Lack of downstream evaluation**. The authors clarified the goals of the downstream evaluation at the outset of Section 5.
  - *Partially addressed*. The clarification does explain the scope of the results. However, there appears to be lacking a summary of the proposed method's performance on established metrics for the downstream tasks.
  - *AC's comment*. Upon review of the LLM Routing results in Section 5.2, we tend to agree with the reviewers' concerns.
    - The results presented fall short of a convincing demonstration that the proposed method leads to better routers; in fact, the authors seem to acknowledge that the setup does not illustrate the value of routing in general, which is a bit confusing (why not pick a setup where routing _does_ help, and show the promise of the method there?).
    - There appears to be an absence of any baseline method.
    - It is common to report a cost-quality tradeoff curve, or summary thereof (e.g., area under the curve, quality-neutral cost).
    - It is advisable to report results on a standard routing benchmark (e.g., RouterBench, SPROUT, EmbedLLM).
    - Some details appear missing: the authors claim _"we
find that prompt embeddings alone are not informative enough to raise performance above that of the best individual model"_, but a precise specification of these results appears missing.

- **Lack of analytical insights**. The authors added a discussion in Section 6 offering a hypothesis as to why latent methods can outperform traditional ones.
  - *Partially addressed*. The provided hypothesis is fairly generic (that the latent information is more fine-grained). At the same time, we understand that a thorough analytical understand can require significantly more work.

**Reviewer Scores:**

- **9WLH**. The reviewers' concerns on training instability and data requirements were at least partially addressed. Thus, we think it possible their score would increase to a 6. However, given their high confidence, and the fact that the concerns may not have been fully addressed, it is also plausible that the score would remain at 4.
- **XxtF**. As the review was mostly positive, and the concerns mostly addressed, we think it plausible for the reviewer to increase their score to an 8; or at least to maintain it at 6.
- **urQp**. Of the reviewers' concerns, perhaps the one about lack of consistent results was the best addressed. The others were only partially addressed. Thus, we do not see it likely for the score to exceed 4, and see it plausible the score would remain at 2.
- **bhBs**. As the review raised a number of areas for improvement, which were partially addressed, we think it likely the score would remain at 6.

---

### Decision · Program_Chairs · 2026-01-26

Reject